# Wrapping glia regulates neuronal signaling speed and precision in the peripheral nervous system of *Drosophila*

Rita Kottmeier[1,3], Jonas Bittern[1,3], Andreas Schoofs[2], Frederieke Scheiwe[1], Till Matzat[1], Michael Pankratz[2] & Christian Klämbt [1✉]

The functionality of the nervous system requires transmission of information along axons with high speed and precision. Conductance velocity depends on axonal diameter whereas signaling precision requires a block of electrical crosstalk between axons, known as ephaptic coupling. Here, we use the peripheral nervous system of *Drosophila* larvae to determine how glia regulates axonal properties. We show that wrapping glial differentiation depends on gap junctions and FGF-signaling. Abnormal glial differentiation affects axonal diameter and conductance velocity and causes mild behavioral phenotypes that can be rescued by a sphingosine-rich diet. Ablation of wrapping glia does not further impair axonal diameter and conductance velocity but causes a prominent locomotion phenotype that cannot be rescued by sphingosine. Moreover, optogenetically evoked locomotor patterns do not depend on conductance speed but require the presence of wrapping glial processes. In conclusion, our data indicate that wrapping glia modulates both speed and precision of neuronal signaling.

[1] Institut für Neuro- und Verhaltensbiologie, Universität Münster, Badestreet 9, 48149 Münster, Germany. [2] LIMES Institute, University of Bonn, Carl Troll Street 31, 53115 Bonn, Germany. [3] These authors contributed equally: Rita Kottmeier, Jonas Bittern. ✉email: klaembt@uni-muenster.de

Any complex nervous system is built by neurons and glia. Neurons collect, compute and transmit information in form of action potentials. The restoration of ion homeostasis as well as the energetically expensive neuron to neuron communication require an enormous amount of energy, which is mostly provided by glia[1–4]. Glial cells are not only indispensable for energy delivery, but perform a large spectrum of additional functions, including physical insulation of the axon to promote faster signal propagation[5–12].

In vertebrates, axonal insulation depends on the axonal caliber. In the peripheral nervous system (PNS), large caliber axons are wrapped by myelin forming Schwann cells which allowed the evolution of saltatory conduction with a 20-fold increase in conduction speed[13,14]. Myelination can be activity dependent and thus conduction velocity can be further regulated according to neuronal activity[15–17]. In contrast, small caliber axons are wrapped in groups by non-myelinating Schwann cells that form Remak fibers[18–20]. Myelinating and non-myelinating Schwann cells share a common lineage and can redifferentiate in the other cell type depending on neuron–glia interaction[21,22]. Interestingly, non-myelinating Schwann cells appear morphologically similar to the wrapping glia found in the *Drosophila* PNS[21,23,24].

Given the generally small size of invertebrates, no evolutionary pressure is expected to promote the development of very fast axonal conductance velocity and thus myelin-like structures. Surprisingly, however, such myelin-like structures can be found in several invertebrates, including shrimps, and copepods which due to their very small size of 200 μm length do not appear to require very fast nerve conduction[25–31]. Indeed, swimming speed in copepods does not correlate with myelination[32]. This suggests that wrapping glial cells perform additional tasks than just the acceleration of axon potential propagation speed[33].

To identify such functions, the larval *Drosophila* PNS provides a powerful model. Peripheral sensory neurons send their axons through the segmental nerves to the ventral nerve cord. At the same time motor neurons project their axons through the segmental nerves to the musculature[34]. The segmental nerves are accompanied by a small set of individually identifiable glial cells which can be placed into three classes according to their morphological and functional characteristics[35–40].

The perineurial and subperineurial glial cells establish the blood-brain barrier[3,23,41,42]. Inside the nerve, peripheral axons are enwrapped by the wrapping glia. Only three to four wrapping glial cells per nerve are specified during embryogenesis[38,39]. They accompany the axons and start to differentiate during the first larval stage. During subsequent larval stages the wrapping glial cells grow and axons are progressively wrapped[23,24]. The differentiation of wrapping glial cells is controlled by a set of transcriptional regulators[43] and receptor tyrosine kinase signaling. The wrapping glial cells of the optic nerve require fibroblast growth factor (FGF)-receptor signaling to wrap around photoreceptor axons[44,45] and wrapping glial cells along the abdominal nerves require EGF-receptor activity and the activating ligand Vein, a *Drosophila* Neuregulin[24]. This process appears evolutionarily conserved since differentiation of myelinating Schwann cells is controlled by the mammalian EGF-receptor and the activating ligand Neuregulin[46–48].

At the end of larval development of *Drosophila* the wrapping glial cell has formed simple glial wraps around axons or small axon bundles[24]. The wrapping glial cells that cover the abdominal nerves can reach up to 2 mm in length, highlighting the need for their efficient metabolic supply. Given the enormous size of the wrapping glia, membrane synthesis is of high relevance. Vesicles required for membrane expansion of wrapping glia are routed via the exocyst pathway to the plasma membrane and respective mutants interfere with wrapping glial differentiation[49]. Moreover,

lack of ceramide synthesis in wrapping glia leads to poor differentiation and a concomitant reduction in conduction velocity[50]. Lack of mactosylceramide, which is generated by the mannosyltransferase Egghead, causes aberrant activation of phosphatidylinositol 3-kinase (PI3K) in peripheral glial cells and might also affect FGF-receptor signaling in wrapping glia[51]. Once differentiated, wrapping glial cells likely participate in metabolic homeostasis[3] and ion homeostasis[52–54].

Here, we address how insulation of axons affects nerve signaling properties. Previously, no specific means to manipulate the peripheral wrapping glia were available. All Gal4 lines known to be expressed in the wrapping glia are also expressed in central glia. We thus establish a Gal4/Gal80 combination which allows to specifically target only the wrapping glial cells. Abnormal wrapping glial differentiation or genetic ablation of wrapping glia cause a reduction in axon caliber and a decrease in conduction velocity. Interestingly, ablation of wrapping glia causes prominent larval locomotor phenotypes, while animals with poorly differentiated wrapping glia show only very mild locomotor phenotypes. To quantify this, we use the coiling phenotype, which increases in animals expressing a dominant negative FGF-receptor and is even higher upon wrapping glia ablation. Interestingly, the coiling phenotype of animals with impaired FGF-receptor activity is rescued to control levels by feeding the larvae with sphingosine, a primary part of sphingolipids found in the plasma membrane. In line with these observations, we find that glial ablation but not poor differentiation of wrapping glia blocks specific behavioral changes evoked by optogenetic means. In summary, our data indicate that the wrapping glial cells guarantee neuronal signaling precision and ensure growth of axons to promote action potential conduction speed.

## Results

**Generation of a wrapping glial driver**. A promoter fragment of the *nrv2* gene is commonly used to target wrapping glia[23,55,56]. We generated a *nrv2-stgGFP* transgene which shows a broad range of activity including many cells in the central nervous system (CNS) (Fig. 1a). This pattern is identical to the pattern directed by the *nrv2-Gal4* driver as seen in the MultiColor FlpOut 2 (MCFO-2) approach[57] (Fig. 1b). Unfortunately, we failed to identify a PNS–glia specific element in the *nrv2* enhancer fragment[58,59]. We therefore screened the Gal4 driver collection generated at Janelia Research Campus, the inSITE collection and many publically available stocks[60–62], but again failed to identify a wrapping glia specific enhancer element.

We next searched for enhancer elements that direct an expression pattern in the CNS as it is observed for *nrv2-Gal4* - but lack the peripheral glia expression domain. This work identified *90C03-Gal4* generated by the FlyLight project[62] which showed no expression in the PNS (Fig. 1c). In the CNS, however, *90C03* and *nrv2* activities are overlapping (Fig. 1d–f). Subsequently, we used the *90C03* enhancer element and established a corresponding Gal80 line by inserting a *90C03-Gal80* transgene into the *attP2* landing site. We then generated a fly strain harboring the combination *nrv2-Gal4, 90C03-Gal80* which resulted in robust expression in wrapping glia with only very little expression outside the nervous system around the pharynx (Fig. 1g, h).

**FGF-receptor activity controls glial wrapping**. To study the impact of wrapping glia on neuronal signaling we then used the above generated tool to specifically manipulate gene activity in these cells. We have previously shown that wrapping glial cells in the developing eye require the activity of the FGF-receptor Heartless (Htl)[44]. To test whether FGF-receptor signaling is also

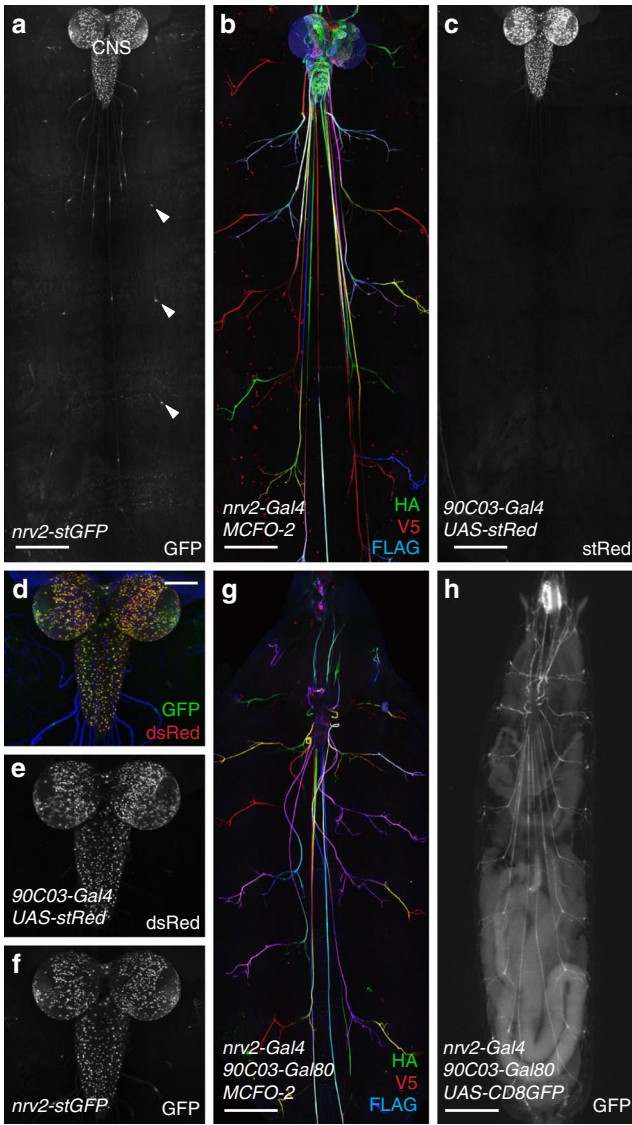

**Fig. 1 Generation of a wrapping glia driver.** Confocal projection of larval filet preparations of the genotypes indicated. Representative images are shown taken from >10 animals analyzed for each genotype. **a** Third instar larva with the genotype [*nrv2-stGFP*]. Broad GFP expression is detected in the CNS. Note the restricted expression in the peripheral nervous system which corresponds to the wrapping glia (arrowheads). **b** Third instar larva of the genotype [*hs-Flp; nrv2-Gal4/MCFO-2*]. *flp* expression of the multicolor FlpOut construct was induced by 1 h 37 °C heat shock during first instar stage. Larvae were stained for HA (green), V5 (red), and FLAG (blue). **c** Same animal as in (**a**). Expression of stRed is observed only in the CNS and no expression is found in the wrapping glia. **d**–**f** Overlay of *nrv2-GFP* (green) and *90C03 > dsRed* (red) expression. Note the complete overlap of dsRed (**e**) and GFP expression in the CNS (**f**). **g** Young third instar larva with the genotype [*hs-Flp; nrv2-Gal4/MCFO-2; 90C03-Gal80*]. *flp* expression was induced by 1 h 37 °C heat shock during first instar stage. Larvae were stained for HA (green), V5 (red), and FLAG (blue). **h** Living third instar larva of the genotype [*nrv2-Gal4, UAS-CD8GFP; 90C03-Gal80/ 90C03-Gal80*]. Note strong expression at the anterior tip of the larva. Scale bars are 250 µm (**a**–**c**, **g**, **h**) and 100 µm (**d**–**f**).

needed during glial differentiation in the peripheral nerve, we expressed a dominant negative Htl construct (*htl^DN*) in wrapping glia. This resulted in larvae with poorly differentiated wrapping glial cells (Fig. 2a, b). In confocal microscopic views the wrapping

glia appeared sometimes thinner, and formed a single line within the axon bundle (Fig. 2a, b; arrows). However, in other nerves or other sections of the same nerve, the wrapping glia appeared to cover a larger region of the nerve bundle (Fig. 2b; arrowheads). Following electron microscopic analysis, the disrupted differentiation of the wrapping glial cell can be clearly seen (Fig. 2c, d). The wrapping glia fails to properly wrap around axons but is still able to extend processes through the nerve (Fig. 2d). The failure to individually wrap axons can be described using the wrapping index ($w_i$) (see "Methods")[24]. In control larvae, the $w_i$ is 18.5 ($n = 26$ nerves, Supplementary Table 1). Upon expression of *htl^DN* in wrapping glial cells, the $w_i$ is about 7.5 (Fig. 2c, d, Fig. 3c; $n = 22$ nerves, $p = 3.5E-11$; Supplementary Table 1).

**Glial FGF-receptor signaling affects axonal diameter.** Concomitantly with the reduced wrapping index we noted that many axons appear smaller in diameter than in wild type nerves of third instar larvae. To quantify the phenotype, we determined the morphology of abdominal nerves about 160 µm distal to the ventral nerve cord as reported previously[24]. We measured the surface area to minimize fixation artifacts and then calculated the axonal radius. The size of 2046 axons in 27 control nerves of 5 larvae was evaluated and we found an average axon radius of 0.194 µm with a characteristic area distribution (Fig. 3a). Upon expression of *htl^DN* in wrapping glia, the axonal diameter is on average reduced by 15% ($r = 0.159$ µm, 2555 axons in 22 nerves from five larvae, $p = 7.6E-26$). When we compared the distribution of axonal sizes between control larvae and larvae with reduced FGF-receptor activity in wrapping glia we found a strong increase in the number of small diameter axons and a decrease in the number of large diameter axons (Fig. 3a, b).

**Behavioral consequences of glial FGF-receptor activity.** In locomotion assays, larvae expressing *htl^DN* in wrapping glia [*nrv2-Gal4/UAS-htl^DN*; *90C03-Gal80/90C03-Gal80*] as well as control animals [*nrv2-Gal4/UAS-GFP*; *90C03-Gal80/90C03-Gal80*] were tracked for 3 min (10 frames per second, ~15 animals per movie, in total ~80 larvae per genotype originating from three independent crosses). The resulting video data were subjected to FIMtrack analysis as described[63–65]. Interestingly, third instar larvae showing such poorly differentiated wrapping glia behaved rather normally. For example, accumulated distance is as in control animals (Fig. 3d). We only observed a difference in bending distribution (Fig. 3e; $p = 8.4E-32$) and a small increase in coiling frequency, a rare behavioral pattern during stop phases, where the bending strength is increased such that the larva turns its head so strong that it touches the tail tip resulting in a coil structure. Whereas in control larvae coiling behavior can be detected in 0.097% of the frames (197,100 frames from 9 movies with 15–20 animals each analyzed), animals expressing *htl^DN* specifically in wrapping glial cells show coiling in 0.774% of all frames (78,900 frames analyzed, $p = 3E-13$). To further quantify this behavior, we analyzed tracks of 300 frames each and plotted the distribution of the coiling frequency. In both conditions, most of the tracks contain 0 frames with a coiled animal. In control animals, only 2% of all tracks contain 1–30 frames showing a coiling pattern. In contrast, 18% of all tracks of larvae with impaired FGF-receptor signaling in wrapping glia show 1–30 frames with coiling in a given 30 s long track. 1.5% of the tracks even show 31–60 frames with coiling, which is never seen in control animals (Fig. 3f). In conclusion, a block of FGF-receptor signaling impairs differentiation of the wrapping glial cells but the residual glial cell processes still allow almost normal neuronal function although axons become considerably smaller.

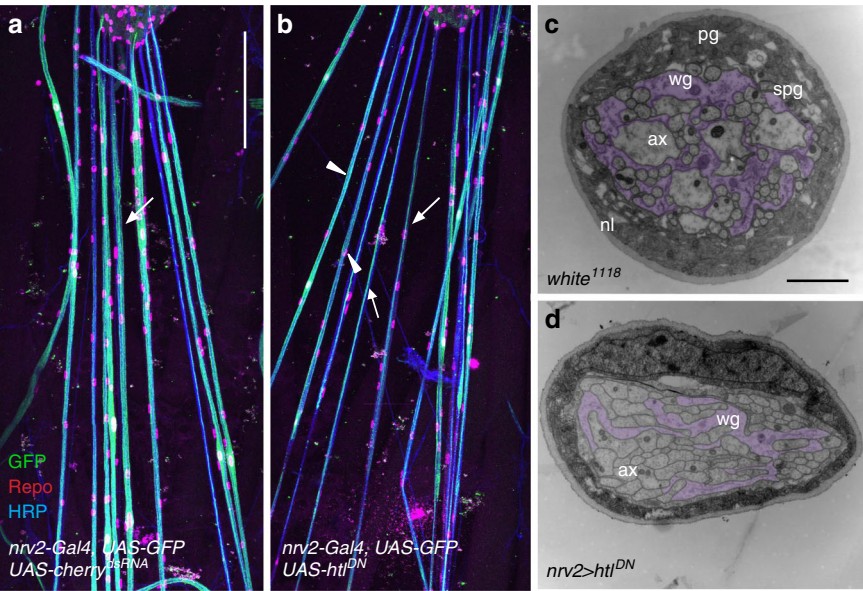

**Fig. 2 FGF-receptor Heartless controls differentiation of the wrapping glia. a, b** Filet preparations of a third instar larvae with a genotypes as indicated stained for Repo expression (red), GFP (green), and HRP (blue). Representative images are shown taken from >10 animals analyzed for each genotype. **b** Note that upon expression of $htl^{DN}$, the wrapping glia appears thinner (arrows). Scale bar is 200 μm. **c, d** Electron microscopic images of segmental nerves taken at about 160 μm distance to the tip of the ventral nerve cord of wandering third instar larvae. Five or more specimens were fixed and embedded in a filleted form. Representative images are shown for each genotype. Scale bar is 2 μm. **c** Cross-section of a control nerve ($white^{1118}$). The nerve is surrounded by a neural lamella (nl). The perineurial glia (pg) and the subperineurial glia (spg) form the blood–brain barrier. The wrapping glia (wg) engulfs all axons (ax). The wrapping glia is highlighted by purple staining. **d** Cross-section of a nerve from an animal of the genotype [*nrv2-Gal4; UAS-htl^{DN}*]. Note that differentiation of the wrapping glia (purple shading) is compromised. A similar phenotype is seen in larvae of the genotype [*nrv2-Gal4/UAS-htl^{DN}; 90C03-Gal80/90C03-Gal80*], see Supplementary Fig. 2).

**Wrapping glial differentiation requires gap junctions**. The observed reduction in axonal diameter suggests that wrapping glial cells metabolically support axons to allow their growth. Metabolic interactions can be either mediated by specific metabolite transporters or by direct coupling of cells through gap junctions[66]. Here, we addressed whether gap junctional coupling is required for wrapping glial growth. *Drosophila* gap junctions are generated by innexins. *innexin1* (*ogre*) and *innexin2* are both expressed by CNS glial cells and are required for viability[67]. To determine the expression of these two innexin genes in wrapping glia we utilized Innexin2 specific antibodies and V5 knockins into the endogenous *ogre* and *innexin2* genes[68]. In peripheral nerves a broad expression of both genes can be detected and both expression patterns show a complete overlap (Fig. 4a). To test whether this is due to neuronal or glial expression we silenced *innexin2* gene activity using RNA interference in the background of a V5 knockin into the *innexin2* gene either in all neurons using *nsyb-Gal4* or in all glial cells using *repo-Gal4*. Upon glial knockdown, no Innexin2$^{V5}$ expression can be seen in abdominal nerves but expression in epidermal cells or tracheae still remains (Fig. 4b). No change in Innexin2$^{V5}$ expression was noted upon pan-neuronal silencing. We then silenced *innexin2$^{V5}$* expression in subsets of peripheral glial cells. However, in this paradigm neither *nrv2-Gal4* which directs expression in wrapping glia nor *moody-Gal4* which directs expression in the subperineurial glia was able to remove *innexin2$^{V5}$* expression completely (Fig. 4c, d). This suggests that *innexin2$^{V5}$* is expressed in all glial cell layers of the segmental nerve, which is in agreement with providing a channel function between the different glial cells.

To further test the functional role of *innexin2* and *ogre* in wrapping glia we analyzed the morphology of the wrapping glia in knockdown experiments. Upon suppression of either innexin gene in the wrapping glia, glial growth is impaired (Fig. 4e–g).

Similarly as noted for upon expression of $htl^{DN}$, knockdown of *ogre* (Fig. 4f) or knockdown of *innexin2* (Fig. 4g) results in poorly differentiated wrapping glial cells. Only thin processes form that fail to grow around the entire nerve bundle. *ogre* knockdown causes mild locomotion defects in third instar larvae. The bending rate was slightly increased and in consequence the accumulated distance per minute was reduced, as animals spent more time in bending compared to control animals (Fig. 4h, i). In addition, *ogre* knockdown in wrapping glia also causes a coiling phenotype. Similar to what we observed for larvae with wrapping glia specific expression of a dominant negative FGF-receptor Heartless, we noted coiling in 0.726% of 77,700 frames analyzed (significantly different when compared to a GFP expressing control: $p = 1.9\text{E} -14$). Also, the distribution of locomotion tracks with increased coiling frequency is comparable to $htl^{DN}$ expression: in 17% of the tracks a coiling is seen in 1–30 frames and in 0.8% of the tracks more than 31 frames showing coiling can be detected in the tracks (Fig. 4j). In conclusion, the suppression of gap junctional coupling causes more pronounced neuronal phenotypes than expressing a dominant negative Heartless FGF-receptor. This supports the notion that wrapping glial differentiation is indeed needed to ensure normal neuronal function.

**Ablation of peripheral wrapping glial cells**. To obtain the strongest possible wrapping glia phenotype, we decided to ablate the wrapping glial cells and induced apoptosis via expression of the gene *head involution defective* (*hid*), a key regulator of developmentally controlled cell death in *Drosophila*[69,70]. To increase the specificity, we included a second *90C03-Gal80* transgene. Expression of Hid in all wrapping glial cells [*nrv2-Gal4/UAS-hid; 90C03-Gal80, UAS-CD8Cherry/90C03-Gal80*] will be referred to as wrapping glia ablation in the following. The

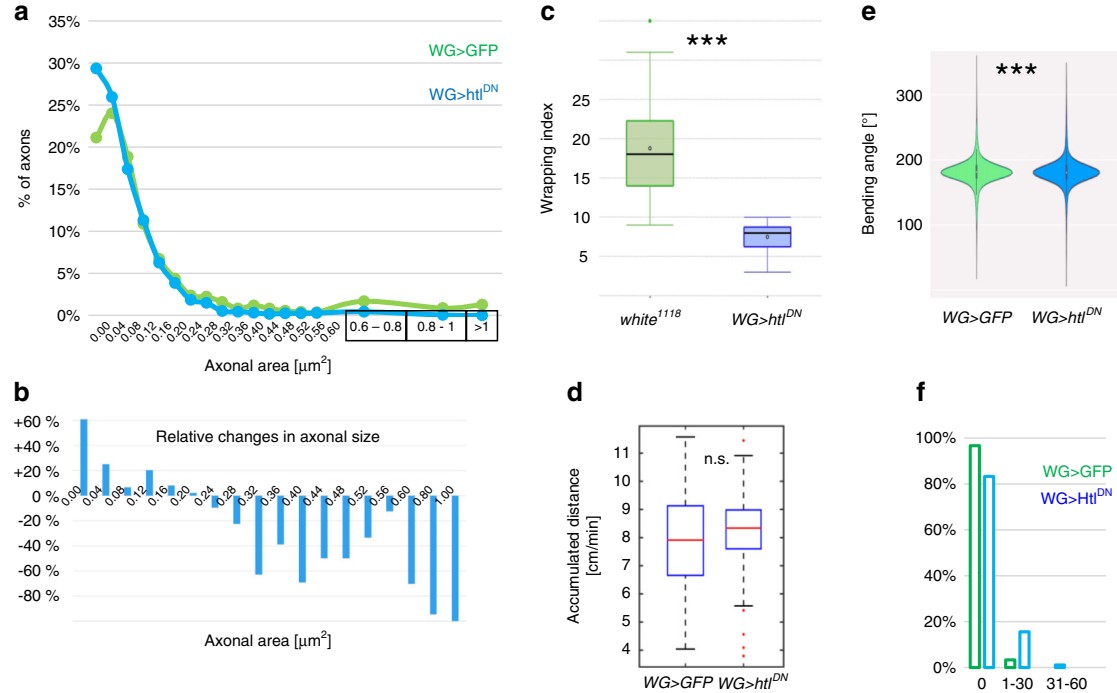

**Fig. 3 Wrapping glia affects neuronal differentiation and behavior. a** Distribution of axonal size in 40 nm$^2$ bins of control larvae ([*nrv2-Gal4, UAS-GFP; 90C03-Gal80/90C03-Gal80*], 2046 axons of 26 nerves) and of larvae expressing *htl$^{DN}$* in wrapping glia ([*nrv2-Gal4/UAS-htl$^{DN}$; 90C03-Gal80/90C03-Gal80*]; 2555 axons of 22 nerves). The number of axons in the indicated size intervals is plotted for control (green) and *htl$^{DN}$* nerves (blue). **b** Same dataset as in (**a**). Relative changes in the number of axons in different axon size classes upon expression of *htl$^{DN}$* compared to control animals. Note the increase in the number of small diameter axons and the reduction of larger caliber axons. **c** The wrapping index of control larvae [*white$^{1118}$*] and larvae expressing *htl$^{DN}$* in the wrapping glia [*nrv2-Gal4; UAS-htl$^{DN}$*]. The wrapping index in control nerves is 18.5 ($n = 27$ nerves). The wrapping index is 7.5 upon *heartless* suppression ($n = 22$ nerves, $p = 3.5E-11$, $t$ test). **d** The accumulated distance per minute is not changed by expression of *htl$^{DN}$* (genotypes as indicated). $n = 68$ control larvae and $n = 71$ larvae expressing *htl$^{DN}$*, $p = 0.85$, Wilcoxon rank-sum test. Box plots in (**c**, **d**) show median (horizontal line), boxes represent the first and third quartile, whiskers show standard deviation, individual points show outliers. **e** Bending angle distribution of control larvae and larvae expressing *htl$^{DN}$* in wrapping glia (genotypes as in (**a**)). A slight difference in the bending behavior can be detected ($p = 8.4E-32$, $t$ test). **f** Quantification of the coiling phenotype. The relative distribution of 300 frames long movement clips ($n = 263$ 30-s long video clips with 300 frames each) with 0, 1–30, or 31–60 frames showing coiling is shown. Green denotes control, blue denotes larvae expressing *htl$^{DN}$* in wrapping glia, genotypes as in (**a**).

expression of *hid* results in an almost complete loss of CD8Cherry expression. No CD8Cherry expression is detected in the PNS during third or second larval stages, indicating death of all wrapping glial cells (Fig. 5a–d). Third instar larvae of this genotype are slightly smaller than control larvae (on average 14% smaller, $n = 10$ larvae, Fig. 6c) and die at the end of pupal development. To determine the branching pattern of the peripheral axons we used the monoclonal antibody 22C10 which labels the *Drosophila* microtubule associated protein 1B, Futsch[71,72]. The expression of Futsch does not differ much between control and wrapping glia ablated animals and the neuronal branching pattern as well as the formation of neuromuscular junctions appears normal (Fig. 5e, f).

**Ablation of wrapping glia causes a reduced axon diameter.** To test how wrapping glia ablation affects axonal morphology, we performed an extensive electron microscopic analysis as described above. We measured the size of 2046 axons in 27 control nerves and the size of 1915 axons in 28 wrapping glia ablated nerves (5 filets were analyzed in every case, size of wrapping glia ablated animals matched control animals). The radius of the axons ranges from about 50 nm to 1 μm (Fig. 5g). In control animals, the average axon radius is 0.194 μm, whereas upon ablation of wrapping glial cells the axonal radius is reduced by 18% to an average axon radius of 0.159 μm ($p = 9.1E-10$). The few larger axons that are detected in wrapping glia ablated nerves are usually found next to the subperineurial glia which engulfs the entire

nerve (Fig. 5h, arrowheads). The reduction in average axonal radius is caused by a strong increase in the number of very small axons and a decrease in the number of larger axons (Fig. 5i, j).

When comparing changes in axonal caliber evoked by wrapping glia ablation to those evoked by expression of *htl$^{DN}$*, only minor differences became apparent (compare Fig. 3a, b to Fig. 5i, j). The proportion of small axons increases and the number of large caliber axons decreases. We thus conclude that glial cells nurture the axons to allow their growth and possibly differentiation. In this respect, it should be noted that during embryonic stages all axons have a rather similar caliber and therefore appear to grow differentially during larval stages[23].

**Ablation of wrapping glial cells affects larval locomotion.** To test possible neuronal deficits, we studied larval locomotion as described above. For larval locomotion assays, 79 wrapping glia ablated larvae originating from three independent crosses [*nrv2-Gal4/UAS-hid; 90C03-Gal80, UAS-CD8-mCherry/90C03-Gal80*] and 91 control animals [*nrv2-Gal4/UAS-GFP; 90C03-Gal80, UAS-CD8-mCherry/90C03-Gal80*] were analyzed. In stark contrast to larvae with wrapping glia specific expression of *htl$^{DN}$*, or larvae where we silenced *ogre*, we noted pronounced behavioral phenotypes in crawling third instar larvae. Upon wrapping glia ablation, larvae do not explore the agar plate as control larvae do (Fig. 6a, b; Supplementary Movies 1 and 2).

Larval locomotion is characterized by a highly stereotypic contraction pattern sweeping across the entire length of the

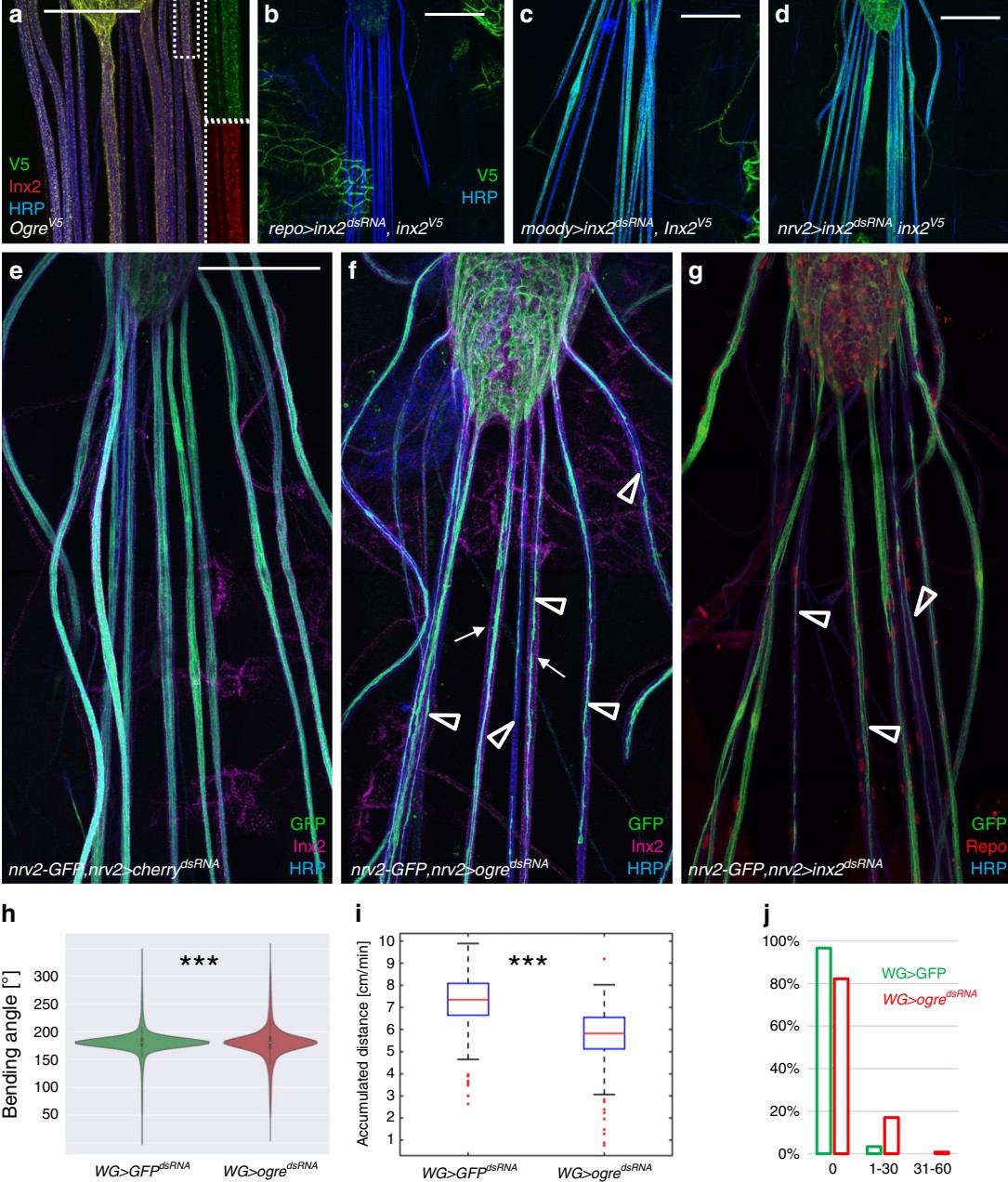

**Fig. 4 Wrapping glial cells require innexins for their growth.** Third instar larval filet preparations expressing endogenously V5-tagged Ogre (**a** green), or endogenously V5-tagged Innexin2 (**b–d** green), or carrying a *nrv2-GFP* fusion (**e–g**, green). HRP (blue) labels neuronal membranes, (**a**, **e–g**) Innexin2 antibody staining is shown in red. More than ten animals were analyzed for each genotype. Representative images are shown. Scale bars are 200 μm. **a** In a control larva, Innexin1 (Ogre, green) and Innexin2 (Inx2, red) show overlapping localization. The boxed area is shown in higher magnification. **b** Upon panglial, RNAi mediated suppression of *innexin2* expression no Inx2 can be detected in nerves. **c** Upon suppression of *innexin2* expression only in the subperineurial glial cells some reduction in Inx2 can be observed. **d** Similar expression patterns are observed upon *innexin2* suppression in wrapping glia. **e** Note that wrapping glial cells cover the entire width of the nerve. **f** Upon knockdown of *ogre* in wrapping glia, the differentiation of wrapping glia is affected (arrowheads). Mainly thin wrapping glial processes can be detected. Inx2 is seen in the peripheral nerves (arrows). **g** A similar phenotype is noted when *innexin2* is suppressed in the wrapping glia (arrowheads). **h** Bending distribution of control [*nrv2-Gal4; 90C03-Gal80/UAS-GFP^dsRNA*] and *ogre* knockdown third instar larvae [*nrv2-Gal4; 90C03-Gal80/UAS-ogre^dsRNA*]. *ogre* knockdown larvae show increased bending ($p = 1.36E{-}25$, t test). **i** The accumulated distance per minute is slightly reduced in *ogre* knockdown animals ($p = 1.6E{-}19$, Wilcoxon rank-sum test, $n = 88$ control larvae and $n = 69$ larvae expressing *ogre^dsRNA*). Box plot shows median (red line), boxes represent the first and third quartile, whiskers show standard deviation, red points show outliers. **j** Same dataset as in (**i**). Quantification of the coiling phenotype. The relative distribution of 300 frames long movement clips (n = 259 30 s long video clips with 300 frames each) with 0, 1–30, or 31–60 frames showing coiling. Green denotes control, red denotes larvae expressing *ogre^dsRNA* in wrapping glia. Genotypes as in (**h**).

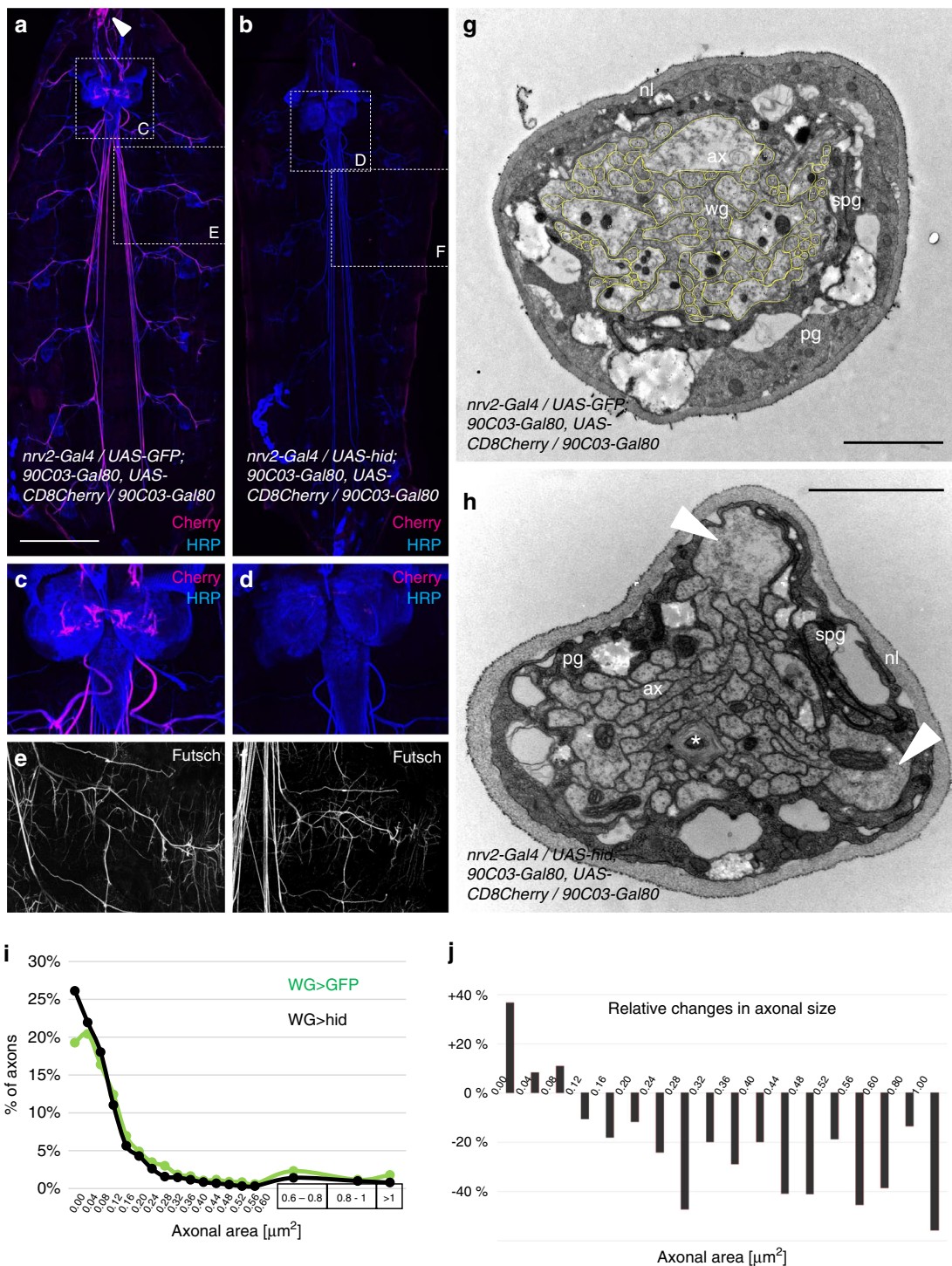

**Fig. 5 Ablation of wrapping glia in abdominal nerves. a** Confocal projection of a control third instar larval filet preparation [*nrv2-Gal4 / UAS-GFP; 90C03-Gal80, UAS-CD8Cherry/90C03-Gal80*]. Expression of CD8Cherry is observed in wrapping glia along abdominal nerves, in a structure close to the mouth hooks (arrowhead) and in few glial cells in the brain (see **c**). Representative images are taken from >10 animals analyzed for each genotype. **b** Wrapping glia ablated third instar larva [*nrv2-Gal4/UAS-hid; 90C03-Gal80, UAS-CD8Cherry/90C03-Gal80*]. Note the absence of CD8Cherry expression (see **d**). Scale bars are 500 µm. **c** Some brain glial cells express CD8Cherry in control larvae. **d** No CD8Cherry expression can be detected following *hid* expression. **e**, **f** Futsch expression as indicated, genotypes as in (**a**, **b**). Note the elaborated branching morphology of peripheral nerves. Upon wrapping glial ablation (**f**) Futsch expression remains unchanged compared to (**e**). **g**, **h** Electron microscopic images of segmental nerves at about 160 µm distance to the ventral nerve cord of wandering third instar larvae. The specimens were fixed and embedded in a filleted form. The size of the different axons was calculated using Fiji (yellow lines). Neural lamella (nl), perineurial glia (pg), subperineurial glia (spg), wrapping glia (wg), axon (ax). Scale bar is 2 µm. Genotypes as in (**a**, **b**). **g** Cross-section of a control nerve. **h** Upon ablation of the wrapping glial cells, only possible remnants of the wrapping glia can be recognized (asterisk). Note that large caliber axons are usually found close to the glial cells of the blood-brain barrier (arrowheads). **i** The number of axons in the indicated size intervals is plotted for control (green) and wrapping glia ablated (black) nerves (*n* = 1901 control axons and 1915 axons in glia ablated nerves, genotypes as in (**a**, **b**). **j** Relative changes in the number of axons in different axon size classes upon ablation of the wrapping glia compared to control animals. Note the increase in the number of small diameter axons and the reduction of larger caliber axons.

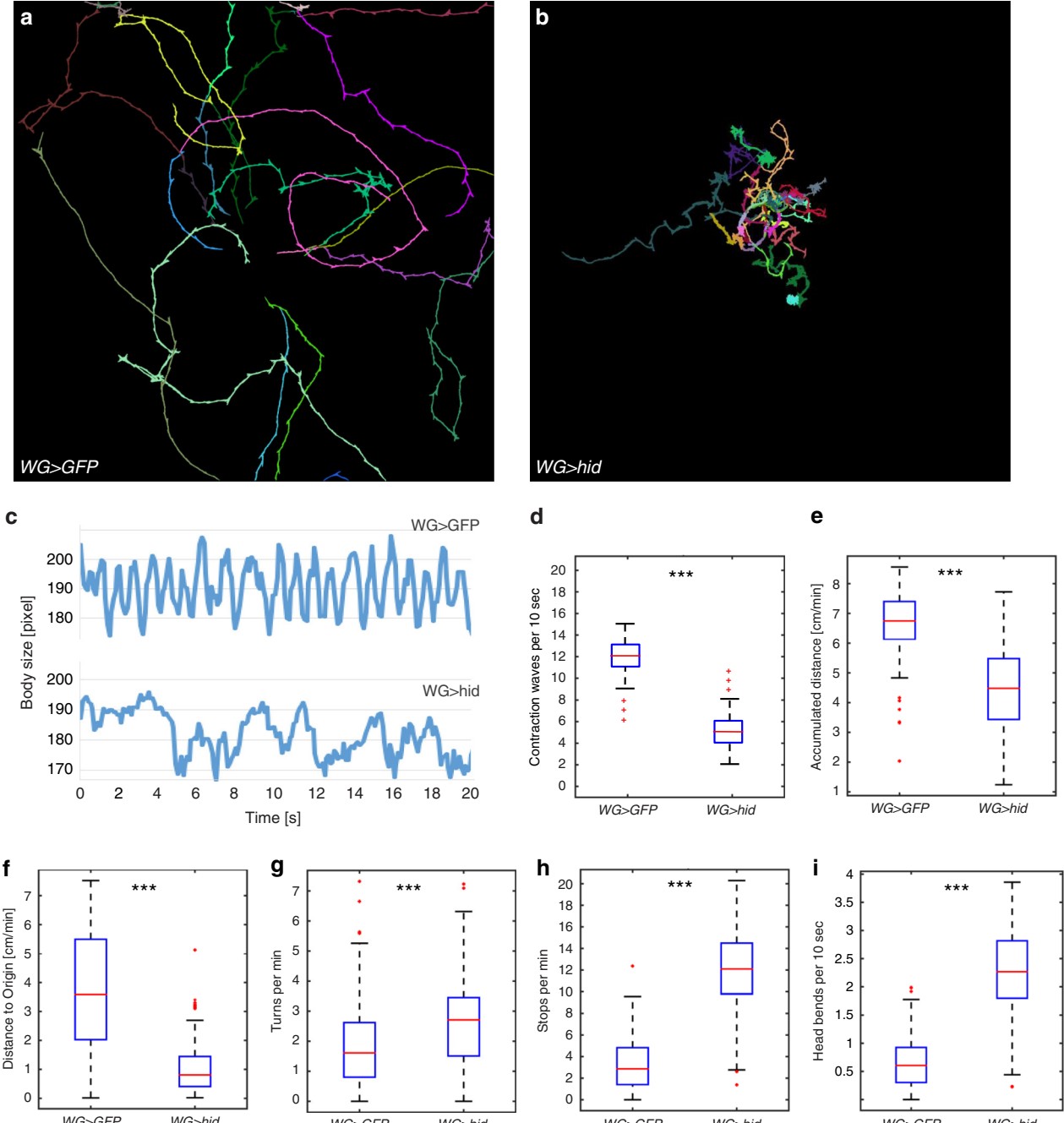

**Fig. 6 Larval locomotion impairment upon wrapping glial ablation. a, b** Representative FIM images. **a** Tracks of 15 control third instar larvae [*nrv2-Gal4/ UAS-GFP; 90C03-Gal80, UAS-CD8Cherry/90C03-Gal80*]. **b** Third instar larvae with ablated wrapping glial cells [*nrv2-Gal4/UAS-hid; 90C03-Gal80, UAS-CD8Cherry/90C03-Gal80*]. Larvae were imaged for 3 min at 10 frames/s. Upon ablation of wrapping glia larvae cover only a small part of the 20 cm × 20 cm arena and perform many turns. **c** Peristaltic body contractions over time. upper panel: control larva, lower panel: wrapping glia ablated larva, genotypes as in (**a, b**). Note the reduction in body size, and reduced and irregular peristaltic waves upon ablation of wrapping glia. **d** The peristalsis frequency is reduced upon wrapping glia ablation (*p* = 9.8E−17). **e–i** Box plots showing quantifications of the larval locomotion parameters. *n* = 91 control larvae, and *n* = 79 wrapping glia ablated larvae. All box plots show median (red line), boxes represent the first and third quartile, whiskers show standard deviation, red points show outliers. **e** Accumulated distance [cm/min], *p* = 9.1E−7. **f** Distance to origin [cm/min], *p* = 1.5E−25. **g** Number of turns/min, *p* = 2.3E−4. **h** Number of stops/min, *p* = 1.6E−35. **i** Number of head bends/10 s *p* = 8.3E−21. **d–g** Statistical analyses: Wilcoxon rank-sum test.

animal from posterior to anterior[73–77]. Upon ablation of wrapping glia, the peristaltic contraction shows a highly aberrant pattern (Fig. 6c). While control larvae show regular contraction patterns with a frequency of about 1.2 peristaltic cycles per second, wrapping glia ablated animals have 0.5 peristaltic cycles per second (*n* = 12 larvae with 10 times 10 s continuous

imaging each, *p* = 9.8E−17, Fig. 6d). Moreover, in wrapping glia ablated animals the shape of the peristaltic curves is generally irregular with many small peaks, indicating that the regular contraction wave is abnormal (Fig. 6c). Further high-resolution analyses showed that in crawling larvae movements of abdominal segments A6–A8, which are most distant to the CNS seem to be

more affected by wrapping glia ablation than segments closer to the CNS (Supplementary Movies 1–3).

Quantification of larval tracks revealed that wrapping glia ablated animals show a decrease in both accumulated distance (Fig. 6e; $p = 9.1E-7$) and distance to origin (Fig. 6f; $p = 1.5E-25$). Interestingly, the distance to origin is much more reduced than the accumulated distance. Further quantification showed a four-fold increase in the number of stops, which are defined as no movements for five frames. In addition, a similar increase in the number of head bends is noted in animals with ablated wrapping glia. Head bends are defined changes in body posture with a larva showing a bending angle of more than 20° over five frames. The increased stop and head bending rates also result in an increase in the number of turns. Here, larvae change direction after head bending (Fig. 6g–i; $p = 2.3E-4$; $1.6E-35$; and $8.3E-21$, respectively).

**Ablation of wrapping glial cells causes a coiling behavior**. Not only the bending rate, but also the bending strength is increased upon wrapping glia ablation. The overall bending distribution was dramatically altered in these animals (Fig. 7a, $p = 1.8E-244$; Fig. 7b, $p = 6.8E-37$, >79 larvae with >230,000 frames analyzed). In contrast to control larvae that move mostly oriented in a straight line, corresponding to a bending angle of 180°, wrapping glia ablated animals show three bending angle peaks. One at 180° corresponding to straight movement and two representing a 25° deviation indicating turning to the left or right. In addition, wrapping glia ablated larvae show an intensive coiling behavior where the head reaches the tail tip (Fig. 7c). Whereas control larvae show a coiling phenotype in only 0.097% of the frames (see above), an almost 16-fold increase in coiling frequency can be noted in wrapping glia ablated larvae. 1.507% of the frames of videos from wrapping glia ablated animals showed coiling behavior (167,100 frames from 8 movies with 15–20 animals each, with 2518 frames showing coiling, $p = 1.2E-12$). This is stronger than what we observed following expression of $htl^{DN}$ specifically in the wrapping glia (0.774% of all frames show coiling see above, Fig. 7d). Only in animals lacking wrapping glial cells we detected 30 s long larval locomotion tracts where more than 60 frames showed coiling (Fig. 7d). In conclusion, these data show that ablation of wrapping glia has a more dramatic consequence than impairing with glial differentiation. This might be due to differential effects on neuronal signaling speed or due to protective effects of glial membranes.

**Wrapping glia controls action potential propagation speed**. To determine the consequence of the different genetic manipulations of the wrapping glia for neuronal conductance velocity we performed electrophysiological measurements. To separate the conductance velocities of sensory and motor axons we established two experimental paradigms.

For motor axons, we utilized an open book preparation of third instar larvae. In such preparations, larval filets show fictive crawling due to a spontaneous activation of the motor program. During fictive crawling, motor neurons driven by the central pattern generator send characteristic trains of action potentials to the muscle which can be recognized in the extracellular recording[78,79] (Supplementary Fig. 1). An electrode was placed at the nerve innervating the seventh or eighth abdominal segment close to the ventral nerve cord to determine the membrane potentials exiting the CNS. A second electrode was placed just anterior to the affected segment (Fig. 8a–c). During a fictive crawling phase, the time interval between individual depolarization events was determined. To calculate the conduction speed,

the time interval between two identifiable depolarization phases was put into relation to the distance between the two electrodes (Fig. 8a, $\Delta d$). The conduction speed of the motor unit is significantly affected by ablation of wrapping glial cells (Fig. 8b, c; $v_{cont\ mot} = 0.196$ m/s [a total of 222 spikes in 16 recordings was analyzed], $v_{abl\ mot} = 0.133$ m/s, [231 spikes/14 recordings]; reduction by 32%, $p = 0.0000125$).

When we analyzed the conductance speed in motor axons following expression of $htl^{DN}$ in wrapping glia we determined a comparable reduction in conduction speed as found in wrapping glia ablated animals (Fig. 8b, c; $v_{hltDN\ mot} = 0.130$ m/s, [178 spikes/10 recordings], $p = 0.00000374$). A reduction of axonal conductance speed is expected since the classical work of Hodgkin and Huxley established that axonal conductance velocity depends on the square root of the axonal radius[80].

To determine the conductance speed in sensory axons, we again selected one of the long nerves innervating the seventh or eighth abdominal segment and severed it close to the ventral nerve cord (Fig. 8d–f). Upon mechanical stimulation of the innervated segment, mechanosensory neurons fire action potentials which propagate along the nerve towards the ventral nerve cord. We calculated the speed by determining the delay of identifiable sensory spikes between the recording sites, knowing the distance between the recording sites. Sensory axons show a 34% reduced conduction speed compared to motor axons (Fig. 8c, f; $v_{cont\ mot} = 0.196$ m/s, $v_{cont\ sens} = 0.129$ m/s, [224 spikes/16 recordings]). Upon wrapping glia ablation, the conduction speed of the sensory unit decreases by over 39% (Fig. 8f; $v_{cont\ sens} = 0.129$ m/s, $v_{abl\ sens} = 0.079$ m/s, [179 spikes/14 recordings], $p = 0.00000587$). The reduction in conduction velocity in sensory nerves of larvae expressing $htl^{DN}$ is significantly less pronounced as induced by wrapping glia ablation (Fig. 8e, f; $v_{hltDN\ sens} = 0.110$ m/s [229 spikes/10 recordings], $p_{(ablation\ vs.\ htlDN)} = 0.009$; $p_{(control\ vs.\ htlDN)} = 0.0229$). Thus, residual glial processes that persist upon expression of $htl^{DN}$ are able to partially sustain axonal conductance speed in sensory axons but are insufficient to maintain conductance speed in motor axons.

**Wrapping glia membrane controls the coiling phenotype**. The coiling phenotype shown by moving third instar larvae appears to be indicative for defects in wrapping glial cell differentiation. Impairment of wrapping glial differentiation results in a medium coiling frequency, whereas wrapping glia ablation causes a strong coiling phenotype. To test whether the locomotor phenotype depends on the presence of wrapping glia membranes, we performed rescue experiments by supplementing the food with sphingosine, a primary part of sphingolipids found in the plasma membrane. Previously, such a food supplementation was shown to promote membrane growth[50]. When we added sphingosine to the food of control larvae, the coiling frequency increased from 0.097 to 0.356% (Fig. 7d, $p = 0.0474$). When we supplemented the food for larvae lacking wrapping glial cells, the coiling frequency increased slightly from 1.507 to 2.166%, ($p = 0.265$, not significant). In contrast, when larvae expressing $htl^{DN}$ in wrapping glia were grown on sphingosine supplemented food, we noted a decrease of the coiling frequency from 0.774 to 0.106% ($p = 0.0001$), which is not distinguishable from the coiling frequency of control animals (0,097%, $p = 0.4209$, not significant). This is also reflected by the lack of tracks with 31–60 frames showing a coiling phenotype (Fig. 7d). Moreover, the addition of sphingosine also rescued the axonal diameter as well as the wrapping index (see Supplementary Table 1, Supplementary Fig. 2). Thus, we conclude that the presence of glial membrane between axons is needed to block coiling.

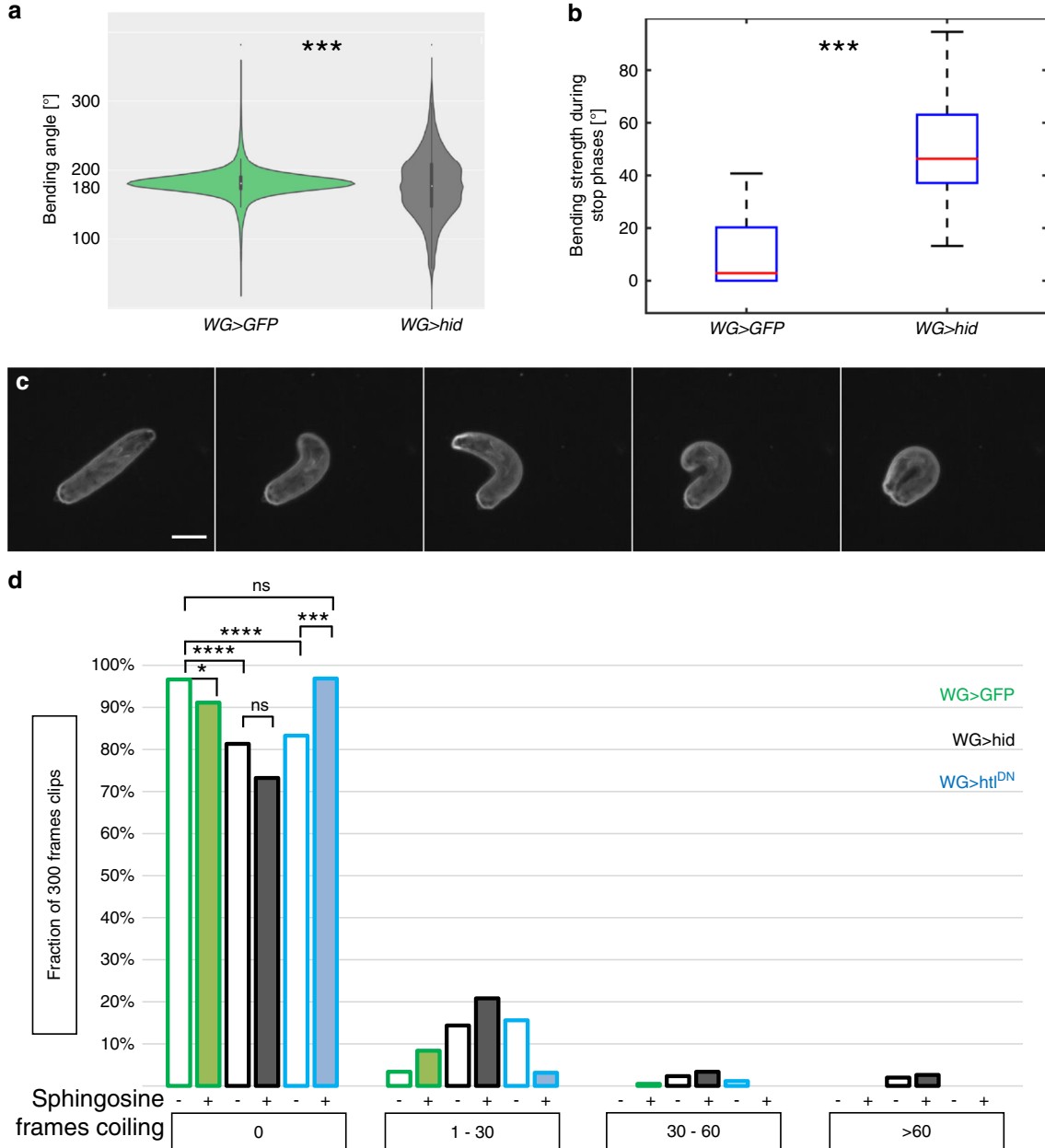

**Fig. 7 Coiling behavior upon wrapping glia ablation. a** Graph showing all bending angles of control [*nrv2-Gal4/UAS-GFP; 90C03-Gal80, UAS-CD8Cherry/ 90C03-Gal80*] and wrapping glia ablated larvae [*nrv2-Gal4/UAS-hid; 90C03-Gal80, UAS-CD8Cherry/90C03-Gal80*] ($p = 1.8E-244$, *t* test). A bending angle of 180° corresponds to a non-bended larva. **b** Box plot shows bending strength [°] during stop phases. Control animals show a median bending angle of around 3°, while Hid expressing animals show a median bending angle of around 47° ($p = 6.8E-37$, Wilcoxon rank-sum test). Genotypes as in (**a**). $n = 91$ control larvae, and $n = 79$ wrapping glia ablated larvae. Box plot shows median (red line), boxes represent the first and third quartile, whiskers show standard deviation, red points show outliers. **c** Five frames of a movie showing coiling of a larva with ablated wrapping glia (genotype as in **a**). Representative example out of 150 coiling events analyzed. Scale bar is 1 mm. **d** Relative changes in coiling behavior. The following genotypes were analyzed: Control (green), wrapping glia ablation (black), genotypes as in (**a**), and larvae expressing *htl*$^{DN}$ in wrapping glia (blue) [*nrv2-Gal4, UAS-htl*$^{DN}$; *90C03-Gal80/90C03-Gal80*]. The proportion of 30 s long video clips (300 frames) showing no coiling, 1–30 frames, 30–60 frames, or more than 60 frames with coiling are plotted. Filled bars indicate animals that were fed with sphingosine. In control larvae coiling is observed in 0.097% of all frames ($n = 657$ video clips of 300 frames each). Upon feeding with sphingosine, coiling frequency increases in control animals to 0.356%, ($n = 203$, $p = 0,0474$). Upon wrapping glia ablation, coiling is noted in 1.507% of all frames ($n = 557$). Upon feeding with sphingosine coiling frequency does not change significantly to 2.166%, ($n = 269$, $p = 0,2653$). Larvae expressing *htl*$^{DN}$ in wrapping glia show coiling in 0.774% of all frames ($n = 263$). Upon feeding with sphingosine, coiling frequency decreases significantly to 0.106% ($n = 223$, $p = 0.0001$) which is not distinguishable from the coiling frequency of control animals (0.097%, $p = 0.4209$). Statistical analyses: Wilcoxon rank-sum test.

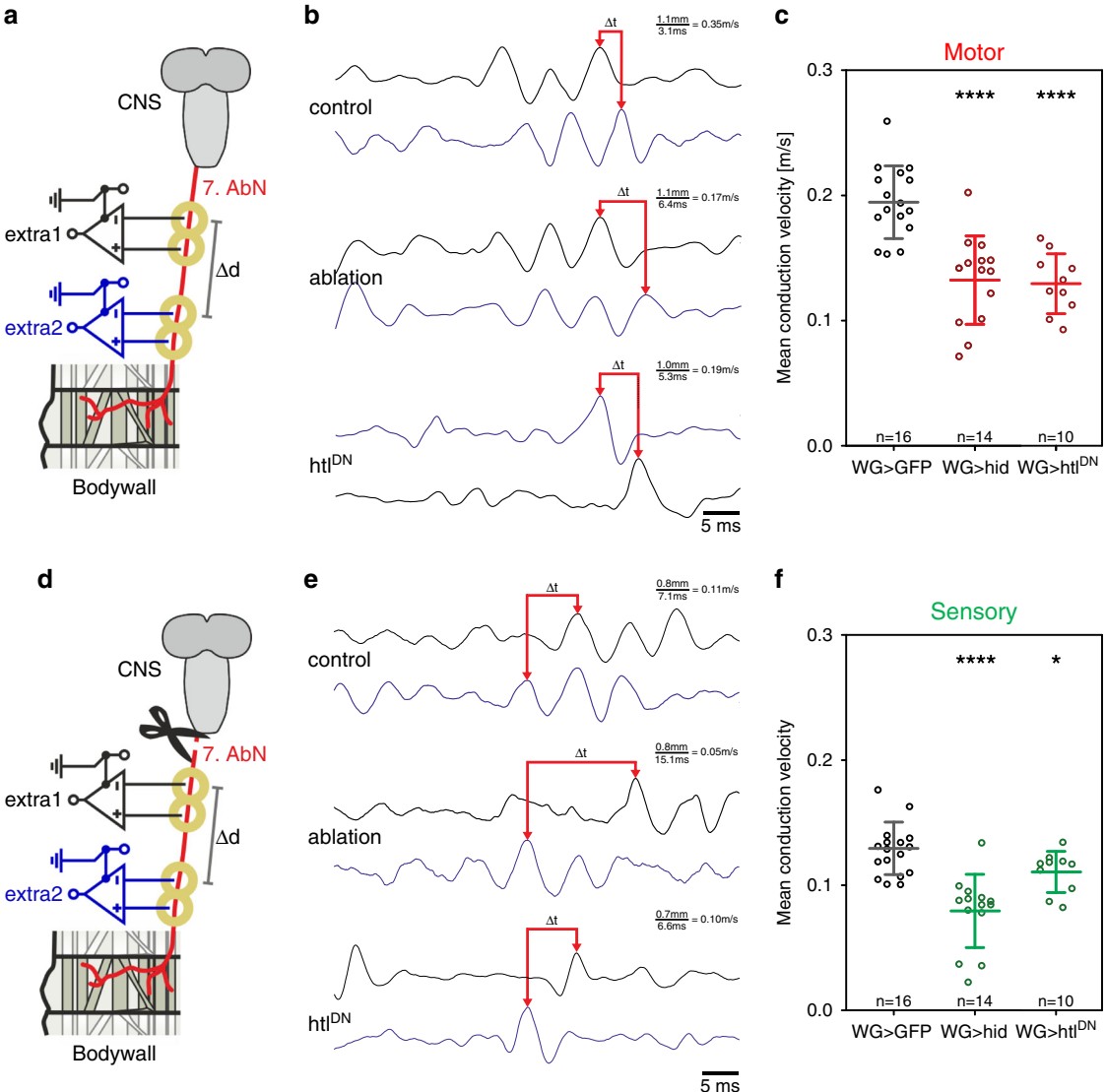

**Fig. 8 Ablation of wrapping glial cells reduces action potential conduction speed. a–c** Conduction speed in motor axons. **a** Schematic representation of the experimental setup. One electrode (extra1) was placed at the seventh or eighth abdominal nerve close to the ventral nerve cord. The membrane potentials exiting the CNS during fictive crawling were recorded by a second electrode (extra2) placed anterior to the affected segment. The time interval ($\Delta t$) between identifiable depolarization phases was put in relation to the distance between the two electrodes ($\Delta d$) to calculate the conduction speed. **b** Exemplary recording traces of the motor unit. Genotypes are as in Fig. 7. **c** The mean conduction speed in control motor axons is 0.196 m/s ($n = 16$ larvae). Upon wrapping glia ablation, the mean conduction speed is 0.133 m/s ($n = 14$ larvae, $p = 0.0000125$). A similar conduction velocity (0.131 m/s, $n = 10$ larvae, $p = 0.00000374$) is observed in motor axons of larvae expressing $htl^{DN}$ in the wrapping glia. **d–f** Conduction speed in sensory axons. **d** To determine the conduction speed in sensory axons, membrane potentials were recorded following mechanical stimulation of the innervated segment (extra1, extra2). The conduction speed was calculated by determining the temporal delay of identifiable sensory spikes between the recording sites and the distance between the recording sites. **e** Exemplary recording traces of the sensory unit. Genotypes are as above. **f** Sensory axons show a 34% reduced conduction speed compared to motor axons (**c**). The mean conduction speed in control sensory axons is 0.129 m/s ($n = 16$ larvae). Upon ablation of the wrapping glia, the mean conduction speed is 0.079 m/s ($n = 14$ larvae, $p = 0.00000587$) corresponding to decrease in conduction speed by 38.5%. In sensory axons of larvae expressing $htl^{DN}$ in the wrapping glia a slightly reduced conduction velocity of 0.110 m/s ($n = 10$ larvae, $p = 0.0229$) is observed. Both plots (**c**, **f**) show the mean and standard deviation. Statistical analyses: Wilcoxon rank-sum test and $t$ test.

**Peripheral wrapping glia controls signaling precision**. The above data suggest that wrapping glial cells provide metabolic support to axons enabling their growth and thus, in direct consequence, provide the means for a fast conductance velocity of action potentials. At the same time wrapping glial cells provide insulation to prevent electrical cross talk between axons. To further test this in a behaving animal, we used the rolling response that is governed by the Goro circuit[81]. Here sensory input of peripheral sensory neurons (mdIV-neurons) is used to trigger the rolling escape reaction involving three layers of

interneurons in the ventral nerve cord and five layers of interneurons in the CNS[81] (Fig. 9). Larval rolling is initiated by the activity of the Goro command neuron. Due to the efforts of the FlyLight consortium enhancer elements are available for all involved neurons[62,81].

When mdIV neurons are activated by optogenetic or thermogenetic means, the animal initiates a rolling response. We expressed the red light sensitive csChrimson in mdIV neurons using the *pickpocket* enhancer (*ppk-LexA*), and recorded the evoked behavioral responses of intact larvae with 100 frames

**Fig. 9 Schematic representation of the Goro circuit. a** Control situation. The rolling response is triggered when a noxious signal is perceived by the mdIV neurons and transmitted via the abdominal nerves to the ventral nerve cord. Here, the mdIV neuron synapses onto the Basin-4 neurons (1) which project to the A05q neurons (2) which innervate the Goro neurons (3) that act as the command neurons orchestrating the activity of motor neurons (4). Additional pathways exist which involve neurons located in the brain hemispheres. **b** Upon ablation of wrapping glial cells, mdIV neuron activation results in ephaptic crosstalk and uncontrolled activation of motor axons. Likewise, upon ablation of wrapping glial cells, Goro neuron activation results in ephaptic crosstalk (flash) and activation of a sensory input affecting the rolling response.

per second. Thereby we were able to obtain a 10 msec time resolution. In control larvae, activation of mdIV neurons by red light triggered a rolling response after 210 msec (median value; $n = 38$ larvae; Fig. 10a, Supplementary Movie 4). When we performed the same optogenetic activation of mdIV neurons in wrapping glia ablated larvae we failed to elicit a rolling response. Instead the larvae showed an uncoordinated seizure phenotype and contracted their musculature without showing a rolling behavior. Importantly, such a reaction was initiated significantly faster and only 80 ms after csChrimson was activated larvae initiated muscle contractions (median value, $n = 28$ larvae; $p = 7E-10$, Fig. 10a, Supplementary Movie 5). When we performed the same experiments using animals expressing $htl^{DN}$ in the wrapping glia, larvae were reacting slightly slower than control animals and importantly were able to initiate rolling ($n = 32$ larvae; $p = 1.9E-5$, Fig. 10a).

In summary, only wrapping glia ablated animals fail to show a coordinated rolling behavior and importantly show fast unspecific contraction responses instead. This indicates that the lack of axonal insulation by glial cell processes results in aberrant activation of motor axons which causes an uncoordinated muscle contraction.

In a next step, we more directly activated the motor neurons by stimulating Goro command neurons via csChrimson expression and red light illumination. As reported[81], this triggers a robust rolling response in control animals after 520 ms (median value, $n = 27$ larvae, Fig. 10b). Interestingly, when we activated csChrimson in Goro neurons of wrapping glia ablated animals, we noted a similarly fast induction of the rolling behavior after 475 ms (median value, $n = 43$ larvae, Fig. 10b). However, when looking at the duration and the coordination of the rolling response we noted highly significant differences between control and wrapping glia ablated larvae. Control larvae showed 5 complete body turns (median value, $n = 23$ larvae). In stark contrast, wrapping glia ablated animals showed a prolonged response to light induced Goro activation and showed 9 full body turns (median value, $n = 33$ larvae, $p = 0.000074$, Fig. 10c, Supplementary Movies 6 and 7).

The rolling response can be either clockwise or counter-clockwise. In control larvae, the rolling orientation does not change after rolling initiation (median = 0 changes in rolling orientation during 5 completed body turns, $n = 27$ larvae). After wrapping glia ablation, the direction of the rolling response frequently changes (median = 4 changes in rolling orientation during 9 completed body turns, $n = 39$ larvae; $p = 0.005$, Fig. 10d). Thus, the rolling direction appears almost randomized for every single turn. When we activated the Goro neurons in the background of an animal expressing a dominant negative

FGF-receptor in the wrapping glia, larvae were reacting slightly slower (Fig. 10b, $n = 26$ larvae, $p = 0.1203$, not significant), showed a similar number of rolls as control larvae (Fig. 10c, $n = 20$ larvae, $p = 0.4839$, not significant) and did not change rolling orientation (Fig. 10d, $n = 20$ larvae, $p = 0.43$, not significant).

In conclusion, in wrapping glia ablated animals, evoked peripheral sensory input into the Goro circuit triggers a fast and rather unspecific activation of motor axons that results in uncoordinated muscle contractions preventing the rolling response. Activation of the Goro command neurons triggers rolling in control and wrapping glia ablated animals. However, upon wrapping glia ablation, activation of motor neurons appears to provide an ectopic activation of neighboring sensory axons resulting in a constant positive feedback which prolongs the rolling response by providing a stimulus for restarting the turning reaction (either clockwise or counter-clockwise).

## Discussion
In *Drosophila*, abdominal peripheral axons are engulfed by only few wrapping glial cells found in each nerve. Using a specific Gal4/Gal80 driver system we found that wrapping glia depend on FGF-signaling and gap junctional coupling. We show that wrapping glial cells are required for radial growth of peripheral axons, suggesting a metabolic support function. Indeed, the lack of nutritive support can be rescued by adding sphingosine to the diet of animals with poorly differentiated wrapping glia. Interestingly, different behavioral consequences are observed in animals with poorly differentiated wrapping glia and those lacking wrapping glia, although both animals show similarly reduced nerve conductance velocities. Locomotor deficits can be restored by a sphingosine enriched diet only in those animals that have poorly differentiated wrapping glia. Further experiments using optogenetically evoked behavior show that wrapping glial cells not only nurture axons but also appear to prevent ephaptic coupling to ensure precise neuronal signaling. Thus, wrapping glia control speed and precision of information transfer in the PNS of *Drosophila*.

In *Drosophila*, all axons start with an almost identical small diameter and during larval stages only few, in particular motor axons, appear to grow to significantly larger sizes[24]. Likewise, axons that are selected for myelination in the mammalian nervous system initially all have very similar sizes and only later, the diameter of myelinated axons is significantly larger than of unmyelinated axon[82,83]. In lampreys, a jawless vertebrate (agnatha), no myelin forms, but still during maturation of the animal axonal calibers increase correlating with axonal ensheathment[84]. This might indicate that the initial recognition of axons to be

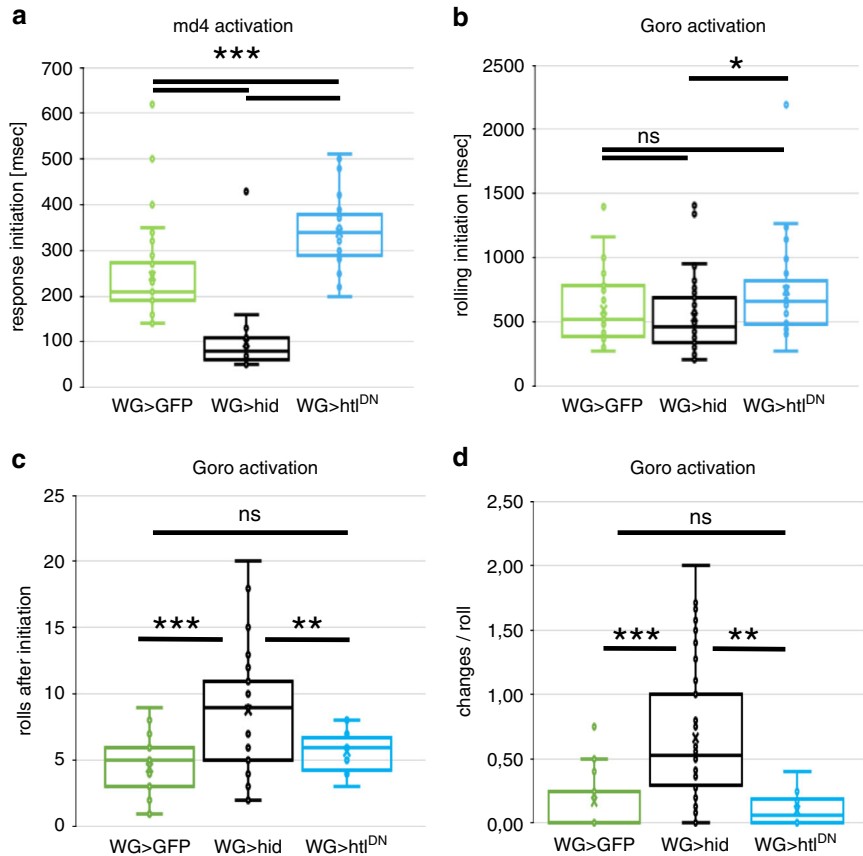

**Fig. 10 Modulation of the Goro circuit. a** Upon optogenetic activation of the mdIV neurons in third instar control larvae [*ppk-LexA, nrv2-Gal4/UAS-GFP; LexAop-csChrimson, nrv2-Gal4/90C03-Gal80*] rolling is induced after 210 ms (*n* = 38 larvae). Upon ablation of the wrapping glia in third instar larvae [*ppk-LexA, nrv2-Gal4/UAS-hid; LexAop-csChrimson, nrv2-Gal4/90C03-Gal80*] no rolling response is induced but larvae cramp after 80 ms (*n* = 28 larvae; *p* = 7E−10). Upon expression of *htl^DN* [*ppk-LexA, nrv2-Gal4/UAS-htl^DN; LexAop-csChrimson, nrv2-Gal4/90C03-Gal80*] a rolling response is initiated as in control but with a delay of 120 ms (*n* = 32 larvae, *p* = 1.9E−5). **b** In control larvae optogenetic activation of the Goro neurons [*Goro-LexA/UAS-GFP; LexAop-csChrimson, nrv2-Gal4/90C03-Gal80*] induces rolling after 475 ms (*n* = 27 larvae). Upon ablation of wrapping glia [*Goro-LexA/UAS-hid; LexAop-csChrimson, nrv2-Gal4/90C03-Gal80*] the rolling response is induced at about the same time (*n* = 43 larvae, *p* = 0.212). A slightly significant delay in the rolling response can be observed when comparing to larvae expressing *htl^DN* [*Goro-LexA, nrv2-Gal4/UAS-htl^DN; LexAop-csChrimson, nrv2-Gal4/90C03-Gal80*] (*n* = 26 larvae, *p* = 0.013). **c** Control larvae show 5 rolls (*n* = 23 larvae, median value). Upon ablation of wrapping glia 9 rolls are induced (median value, *n* = 33 larvae, *p* = 0.000074). Larvae expressing *htl^DN* in wrapping glia show 6 rolls which is not significantly different from the control (*n* = 20 larvae, *p* = 0.48). Genotypes as in (**b**). **d** The orientation of rolling upon Goro activation changes when wrapping glia are ablated but not when wrapping glia differentiation is impaired. Control larvae (*n* = 27) and larvae expressing *htl^DN* (*n* = 20) do not change orientation of rolling reaction (*p* = 0.43). In contrast upon wrapping glia ablation larvae change orientation during every other roll (*n* = 39 larvae, *p* = 0.005). Genotypes as in (**b**). Box plots: median (horizontal line), boxes represent first and third quartile, whiskers show standard deviation, individual points show outliers. All statistical analyses: *t* test.

myelinated does not depend on size but that rather those axons could be selected that show most robust neuronal activity (e.g., motor axons). In fact, activity-dependent myelin plasticity is documented[10,85]. Therefore, one might conclude that of initially similarly sized axons, the more active ones are selected by the glia and subsequent wrapping then promotes axonal growth. The finding that the reduced axonal diameter associated with poorly differentiated wrapping glia in the *Drosophila* larva can be rescued by the addition of extra sphingosine to the food supports the notion that glial cells nurture axons to allow their growth.

In the mammalian PNS differential axonal caliber has long been recognized as an important denominator of myelination[82,83]. Whereas large caliber axons are associated with myelinating Schwann cells, small caliber axons are associated with non-myelinating Schwann cells that form Remak fibers[18–20]. Many of the axons in Remak bundles are sensory C-fibers which transmit pain information[21,86]. Differentiation of the non-myelinating Schwann cells is in part regulated by GABA-B-receptor signaling[87]. Schwann cell specific knock-out mouse mutants show an

increase in the number of small unmyelinated fibers. Interestingly, such mice show an increased sensitivity to pain (hyperalgesia) and, as also seen upon deletion of the LDL receptor-related protein-1, show pain reactions to stimuli that normally do not elicit pain (allodynia)[87,88]. Neuropathic pain due to poorly differentiated non-myelinating Schwann cells might be indicative for ephaptic coupling of closely contacting axons similar to what we reported here.

Non-myelinating Schwann cells, which share a common lineage with myelinating Schwann cells[22,89,90], are morphologically similar to the wrapping glia that engulfs peripheral axons in *Drosophila*[21,23,24]. Moreover, similar molecular processes appear to operate in flies and mammals. In mice, deletion of FGF-receptor signaling specifically in non-myelinating Schwann cells leads to abnormally and smaller appearing axons[91]. Likewise, differentiation of non-myelinating Schwann cells in mice requires neuropathy target esterase while glial loss of the *Drosophila* homolog *swiss cheese* causes incomplete glial wrapping as well as locomotion defects[92,93].

Axonal ensheathment by glial cells establishes a metabolic barrier and thus calls for intensive metabolic neuron-glia coupling which has been found both in invertebrates and in vertebrates[3,7–10]. Possibly, the more metabolic support is provided, the more glial membranes insulate an axon, and at the same time the growth of the axon is fostered. This is supported by our finding that the addition of sphingosine to the food rescued glial complexity and axonal growth in larvae with impaired wrapping glial differentiation but not in those that lacked the wrapping glia. The coiling phenotype that is rarely seen in control larvae, is increased in animals with impaired wrapping glia morphology (either caused by expression of $htl^{DN}$ or by silencing of $ogre$) and even more in animals lacking wrapping glia. Again, the addition of sphingosine to the diet is able to rescue behavioral deficits only in those larvae with impaired wrapping glial differentiation but not in those that lacked wrapping glia, suggesting that the increase of glial complexity ensures neuronal signaling precision.

It is long known that the increase of axonal diameter directly leads to an increase in nerve conduction velocity[28,80,94]. Glial cells nurturing the axon to allow growth to a larger diameter thus participate in the regulation of conduction speed. Interestingly, disruption of wrapping glia differentiation as well as ablation of the wrapping glia both resulted in significantly reduced conduction velocity along motor axons. In contrast, conductance velocity in sensory axons is more strongly affected by wrapping glia ablation than by mal-differentiated wrapping glia. Thus, we speculate that wrapping glia may affect conductance velocity not only by controlling the axonal diameter but also by regulating the functioning of voltage gated ion channels.

Interestingly, larvae with mal-differentiated wrapping glia show a relatively normal locomotion pattern, whereas larvae lacking wrapping glia show pronounced locomotion phenotypes. In addition, only the lack of wrapping glia has an effect on the optogenetically evoked rolling response. Thus, the question remains how the mere presence of wrapping glial cells influences locomotor behavior. One known function of wrapping glia is to control water and ion homeostasis, which in turn influences neuronal activity. Disruption of the salt-inducible kinase 3 (SIK2), which regulates ion transporter expression in wrapping glia or the Serine/Threonine kinase Fray (the orthologue of the mammalian PASK), which regulates the $Na^+$-$K^+$-$Cl^-$ cotransporter Ncc69, or the $K^+$-$Cl^-$ cotransporter Kcc, all result in increased interstitial $K^+$ concentration resulting in hyperexcitability[52,53,95–97]. In addition, the elevated $K^+$ levels also cause water influx and edema in the affected nerves[52,53,95–97]. However, since we never observed such edema following wrapping glia ablation, alternative mechanisms are likely to explain how the lack of wrapping glia accounts for the observed behavioral phenotypes.

Possibly, the observed locomotion phenotypes indicate a role of wrapping glia in suppressing unwanted electrical crosstalk between individual axons in peripheral nerves, known as ephaptic coupling. Ephaptic coupling generates noise in neuronal information transmission and thus must be blocked between most axons to avoid unwanted neuronal stimulation. However, ephaptic coupling is beneficial during synchronous neuronal activity[98–100]. Unfortunately, the small size of the axons in the abdominal nerves and the lack of knowledge on which axons are neighboring each other precludes direct electrophysiological studies to demonstrate ephaptic coupling. We therefore evoked the activity of defined neurons and determined the physiological output. Upon optogenetic activation of the mdIV neurons, control larvae exhibit a prominent rolling behavior[81]. Only when wrapping glial cells are ablated, no rolling response is initiated but instead a fast seizure like reaction can be observed (Figs. 9 and 10).

In a control 2 mm long peripheral nerve it takes 15.5 ms to deliver a sensory stimulus to the CNS and 10.2 ms to deliver the information from the CNS to the neuromuscular junction along the motor axon. In a wrapping glia ablated nerve, it takes 25.3 ms to deliver the signal to the CNS and 15 ms to send the motor information. Moreover, each synaptic transmission event in the CNS would account for an additional 10 ms which would add 10–50 ms depending on the number of synaptic contacts. Given the unknown kinetics of action potential formation in the mdIV neurons, ephaptic coupling between sensory and motor axons therefore appears to be a likely underlying mechanism explaining the fast reaction time of 80 ms. Residual glial cell processes, as we find them in larvae expressing a dominant negative FGF-receptor specifically in the wrapping glia, are sufficient to block ephaptic coupling, allowing a relatively normal rolling response in contrast to what we observed following ablation of these glial cells.

Interestingly, when we triggered the activation of the Goro command neuron, a rolling response is activated in both control and wrapping glia ablated animals, but wrapping glia ablated animals show almost twice as many rolls and, moreover, change the rolling direction almost every single turn (Figs. 9 and 10). This demonstrates that wrapping glia ablated animals are in principle able to perform the rolling response. When a rolling command is given and motor axons are active, sensory neurons are likely activated by ephaptic processes which on the one hand provides a feedback stimulus that prolongs the rolling response and on the other hand resets the rolling direction with every turn.

Therefore, we suggest the following model underlying neuron-glia interaction in the $Drosophila$ PNS. Motor neurons are active from beginning onwards as the larva needs to crawl as soon it leaves the egg case. Possibly high neuronal activity helps to attract glial processes which subsequently start to ensheath these axons. The axon–glia contact also triggers an increased metabolic rate in the wrapping glia which subsequently metabolically support the axons, resulting in axonal growth. Concomitantly, the glia provides signals for proper axonal differentiation and separates axons from neighboring ones. This separation then results in a reduction of ephaptic coupling and thus electric noise during neuronal signal transmission. This might represent a further evolutionary advantage towards the development of myelin[33].

## Methods

**Drosophila genetics**. All *Drosophila* work was conducted according to standard procedures. The following fly strains were used in this study: *90C03-Gal4*, *MCFO-2*[57,62], *repo-Gal4*, *nrv2-Gal4*, *UAS-CD8Cherry*, *UAS-GFP*, *UAS-hid*, *R27H06-LexA* (=*ppk-LexA*), *16E11-LexA* (=*Goro-LexA*), and *LexAop-csChrimson* (Bloomington), *moody-Gal4*[23]. *UAS-heartless*[dsRNA], *innexin2*[dsRNA] (VDRC Vienna, KK102194), *ogre*[dsRNA] (BL44048), *UAS-heartless*[DN101]. *90C03-Gal80* was generated by inserting a *90C03* enhancer fragment[61] (primer 1: GAGTTATGAGCTAGATCCGGTCGTA; primer 2: CTCGCTCGGAGGAAACTGTTGACTG) upstream of the Gal80 sequence. All crosses were performed at 25 °C. Induction of *hs-flp* expression for multicolor labeling was done by placing first instar larvae at 37 °C for one hour as described[57]. Supplementation of sphingosine (Sigma S7049) to the food was performed as described[50]. Larvae were kept on sphingosine food from embryonic stages onwards.

**FIM imaging**. Larval locomotion was analyzed using the frustrated total internal reflection-based imaging method FIM[102,103]. Recordings of about 15 larvae each were performed at room temperature. Larvae were recorded for 3 min with 10 frames per second. Tracking data were obtained using FIMTrack (http://fim.uni-muenster.de[64]) and analyzed using FIManalytics v0.1.1.2 and Excel. Statistical analyses were performed with MatLab (MathWorks, Wilcoxon rank-sum tests). Accumulated distance describes the total length of larval trajectories per minute. Distance to origin describes the distance the larvae moved away from the spot they were placed normalized per minute. A stop is defined as no movement for at least five frames. The number of stops per minute and animal is given. A head bend is defined when a larva shows a bending angle of at least 20° for at least 5 frames. The number of head bends per 10 s is given. In bending distribution plots all deviations from an 180° body axis are show. Totally, 180° indicates no turn and straight appearance. Turns are defined as a stop lasting at least 4 frames followed by a body bending of at least 20° and subsequent forward locomotion towards the new direction for at least 20 frames. Peristalsis frequency gives the number of body contractions as defined by rhythmic alterations in body size per 10 s. Peristalsis efficacy defines the distance a larva crawls per peristaltic wave.

**Immunohistochemistry**. For confocal analyses, we dissected at least six to ten animals. For fixation larvae were opened at the dorsal midline and stretched out with four needles, fixed in PBS 4% PFA for 20 min at room temperature and then washed with three quick buffer exchanges and three 20 min long washing steps. Tissues were blocked for immunohistochemistry with 10% goat serum for 1 h as described[43]. The following antibodies were used: Anti-V5 (1:500, Invitrogen, R96025); anti-Flag (1:1000; Novus biologicals, NBP1-06712); anti-HA (1:1000; Covance, MMS-101P 901503); anti-HRP-DyLight™649 (1:500; Dianova, 123-165-021); anti-GFP (1:1000; Invitrogen, A6455); anti dsRed (1:1000, Clontech Labs 3P 632496), anti-Innexin2 (1:100[67]) anti-Futsch (1:5, mAb 22C10[71,72]), conjugated secondary antibodies (all 1:1000, anti-mouse 488, A10680; anti-mouse 568, A11031; anti-rabbit 488, A11008; anti-rabbit 568, A11011; anti guinea pig 647, 1903515; anti-rat 647, A21247; all Invitrogen). All specimens were analyzed using a Zeiss 710 or 880 LSM; orthogonal sections were taken using the Zeiss LSM imaging software.

**Electron microscopic analysis**. For electron microscopy analyses larvae were dissected and fixed as filets in 4% paraformaldehyde (PFA) and 0.5% glutaraldehyde at 4 °C overnight. Fixation and embedding of larval filets was done as described before[24,104]. Ultra-thin sections of segmental nerves about 160 µm distant from the tip of the ventral nerve cord were obtained using a Leica Ultramicrotome UC7 and were stained with 2% uranyl acetate for 30 min at room temperature in the dark. Ultrathin sections were directly imaged with a Zeiss EM900 using a SIS Morada digital camera.

**Electrophysiology**. For recordings, third instar larvae (98 h ± 2h) were washed and dissected as described[65]. To measure the conduction velocity of neuronal units, double extracellular recordings of the seventh or eight abdominal nerve were performed. Neural activity was measured by differential extracellular recordings with a preamplifier (Model MA103, electronics lab University of Cologne) connected to a four-channel amplifier signal conditioner (Model MA102, electronics lab University of Cologne). All recorded signals were amplified (amplification factor: 5,000) and filtered (bandpass: 0.1–3 kHz). The recordings were sampled at 20 kHz. Recording electrodes were made of silver wire (diameter: 50 µm, Goodfellow). Data were acquired with a Power 1401mk2 A/D board (Cambridge Electronic Design) and Spike2 (version 7) software (Cambridge Electronic Design).

**Experimental design and statistical analyses**. To determine the conduction velocity of neuronal units in the seventh or eighth abdominal nerve, we used the distance between the two different electrodes of the double recording. After each double recording an image of the recording sites was taken and the distance was measured by using the software Fiji (ImageJ version 1.52b). The delay of a specific neuronal unit in the double recording was determined by the usage of a custommade analysis-script in Spike2 (https://github.com/Pankratz-Lab/Spike2-Scripts). The specificity of the analyzed neuronal units over the series of recordings was ensured by spike sorting, an internal analysis tool in Spike2. The conduction velocity was calculated as distance/delay-quotient. For each double recording the mean conduction velocity for a motor and sensory unit was determined. Outliers were determined using Grubb's test and removed from the analysis. The statistical analysis of the data were done using SigmaPlot (version 12). Normal distribution was tested using the Shapiro Wilk test. The significance was tested by using a Wilcoxon rank-sum test (nonparametrical) and $t$ test (parametrical). In control larvae 222 spikes were analyzed for motor axons and 224 spikes were analyzed for sensory axons. In wrapping glia ablated larvae 231 spikes were analyzed for the motor component and 179 spike were analyzed for the sensory system. For larvae with impaired FGF-receptor signaling we analyzed 179 spikes for motor axons and 229 spikes for the sensory axons. $p$ Values are given as two-tailed $p$ values.

For statistical analyses of FIM and EM data the standard nonparametrical Wilcoxon rank-sum test and the standard parametrical $t$ test were performed using MatLab as described[64,65]. Normal distribution of data was tested using the Shapiro Wilk test and the statistical test was chosen accordingly. To calculate the wrapping index the number of individual wrapped axons or axon fascicles is put into relation to the number of all axons. A wrapping index of 100 implies, that every single axon of the nerve is individually wrapped. All nerves that contained less than 76 or more than 82 axons were not included in the statistical analysis. Significance was determined using the Student's $t$ test.

**Reporting summary**. Further information on research design is available in the Nature Research Reporting Summary linked to this article.

## Data availability

All fly stocks generated for this article are available from the corresponding author (C.K.) upon request. The Spike2 analysis script is available through github (https://github.com/Pankratz-Lab/Spike2-Scripts).

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

## Acknowledgements

We are thankful to P. Plelan for sending anti-Innexin2 antibodies, H. Kranenburg for advice throughout the project, A. Volkenhoff for help during the innexin analysis, F. Sieglitz and S. Thomas who were involved in the early steps of this project, S. Luschnig, S. Rumpf, S. Schirmeier, and B. Zalc and all lab members for fruitful discussions. We are indebted to E. Naffin for help with the fly work. This work was supported through grants of the DFG (SFB 1348 B5; Kl588/27-1).

## Author contributions

R.K., J.B., and C.K. designed the experiments. A.S and M.P. performed the electrophysiological experiments, F.S. and T.M. performed the TEM analysis. R.K. and J.B. performed all other experiments. The paper was written by C.K. and commented by all the authors.

## Funding

## Competing interests

The authors declare no competing interests.
