## [Peer Review File · Nature Communications]

Reviewers' Comments:

Reviewer #1:

Remarks to the Author:

In this manuscript Klämbt and colleagues report on the role of wrapping glia in the *Drosophila* using a combination of genetic, optogenetic and electrophysiological approaches. While wrapping glia tightly ensheath axons they do not form multilamellar membrane sheaths as the vertebrate oligodendroglial myelin sheaths. Here, the authors examine the hypothesis that wrapping glia, in homology to vertebrate myelin, contribute to conduction velocity and/or metabolic supply. The genetic approaches to ablate wrapping glia (based on expressing *Hid* and increasing specificity with the 90C03-Gal80 transgene combination), the whole-body imaging as well as the behavioral analyses are convincing. The questions are valid, timely and interesting. However, a number of claims made in the present manuscript version are speculative and the evidence that wrapping glia contributes to metabolic supply is absent.

Major points.

A) The main finding is that the authors (p. 4) "... identified behavioral changes that cannot be simply accounted to reduction action potential conduction speed". What is the evidence that reduced conduction velocity is insufficient? Which experiments are supporting those claims? It is for example unclear how equation 1 (p. 17) is applied and used by the authors. Which numbers are thought to be used and what would be the predicted mean velocity change? Computational studies of myelin have shown that even the presence of a single membrane wrap, increasing the extracellular resistance and reducing axial leak, can support velocity increases. How do the authors distinguish between an electrical insulating effect and axon core diameter reduction?

B) There is no direct evidence for a role of metabolic supply. The conjecture that an average axon diameter reduction in the absence of wrapping glia is evidence for metabolic supply is correlative. Axon diameters are in part dependent on various neuron-glia interaction partners including Neuregulin. If the homology with vertebrate myelin is used the authors will need to demonstrate knocking out lactate or alanine uptake or other metabolic transporters mimics the findings and/or contribute to the phenotypes. Alternatively, experiments will need to demonstrate that morphological phenotype can be rescued, such as e.g. adding sphingosine to the diet (Ghosh et al., *PLoS Gen.* 2013).

C) The terminology and use of electrophysiological readouts such as "ephaptic coupling" is confusing. Ephaptic coupling refers to the finite size and resistance of the extracellular space, causing electrical interference from an action potential in one fiber to the other. In the absence of wrapping glia and the increased extracellular space it may be even envisioned that electrical interactions will diminish. On the other hand, as the authors also mention in the Discussion, "Ephaptic coupling can be beneficial during synchronous neuronal activity". The authors interpret optogenetically evoked peripheral sensory input into the Goro circuit as seizures and evidence for ephaptic coupling. This is insufficient. It is also unclear why this would inform us about "precision" of action potential propagation or whether this is evidence for "ephaptic coupling". To demonstrate ephaptic coupling with electrophysiological means is indeed difficult. However, I would recommend the authors attempt to provide electrophysiological evidence for a role of wrapping glia in temporal precision, which is doable. The authors may want for example to record many trials with the extracellular responses (better using suction electrodes) and compare quantitative measures like jitter (i.e. the variation following a precisely triggered pulses).

Minor points.

p. 3 The Introduction fails to provide a rigorous in-depth overview of wrapping glia in *Drosophila*

but instead superficially describes aspects like "training of the brain" or the "allowance of dinosaurs". This is interesting for an Opinion article or Review on myelin but has no place in an introduction into wrapping glia in the fly. This reviewer was missing an introduction about the genes used to ablate wrapping glia. A more concise Introduction into those would be helpful. Introduce the genetics instead of a generic statement like "We established a tool".

p. 3 The sentence that "This suggests that wrapping glial cells perform additional tasks" I was not able to follow. How are the two preceding sentences about myelination in shrimps and copepods supporting this argument?

p.3. "Bundles m" ?

Figure 7. What do the multiple depolarizations in the extracellular recordings represent? A more detailed description would be useful. A particular concern is which waves are selected for calculating the temporal delay? In addition, how many trials are recorded? How robust are the differences?

For the majority of the data reported in this study detailed information about supporting statistical approaches are lacking. Just one example: "upon ablation of wrapping glial cells axon diameters and the corresponding axons areas are reduced" (p. 9). No numbers, standard deviations nor supporting statistics are reported. Neither in the Figure legends. Proper scientific reporting for all statistical comparisons and claims needs to be included.

In the methods "double-paired t-test" is mentioned which is unknown to this reviewer.

Reviewer #2:

Remarks to the Author:

In the current manuscript, the authors examine the role of wrapping glia in *Drosophila* larvae as a step towards myelinating glia in vertebrates. The authors generate a new genetic toolkit to manipulate wrapping glia specifically and show that reductions in their function cause only mild defects in larvae. In contrast, complete ablation of the wrapping glia leads to smaller diameter axons and a host of defective behaviors, including reduced motor and peripheral axon conduction speeds. The authors suggest their data support an increase in ephaptic coupling (increased electrical cross-talk between axons) based on a behavioral rolling response observed in a sensory to motor circuit, but this conclusion has the weakest support of the other claims. Overall, the authors generate a nice tool to study wrapping glia function as a precursor of the evolution of myelinating glia and report interesting functions for this glial population in regulating axonal size and action potential conduction velocity.

Minor Comments:

1) Images shown in Figure 2B and in Figure 3F/G are used to suggest that manipulating wrapping glia development makes the glia "thinner." However, the EM images of these manipulations show the wrapping glia extending throughout the nerve as seen in controls. The only EM deficit appears to be failure of the glia to fully wrap around axons. Why does this ultrastructural change result in a thinner appearance by confocal microscopy? How to directly compare the EM and confocal images is a bit unclear. Indeed, understanding what is happening in the early manipulations of the wrapping glia before the authors ablate them is a bit tricky.

2) Figure 7 shows wrapping glia are important for regulating axonal conduction velocity. Whether this is directly caused by leaky conduction, or indirectly by failure to support axonal maturation is not clear. The authors favor reduced axon diameter as the explanation, but it seems there are other possibilities here that can't be ruled out. It's also difficult for me to determine how the data

in Fig. 7B was generated – the two extracellular recordings don't always match up well, even in their presumed best sample traces they show – how are the authors sure their picking the same action potential to measure at the more distal electrode? This is some of the most critical data for the paper.

3) Figure 8 is a nice experimental test of a larger sensory to motor circuit, but experimental interpretations seem open to some guesswork due to the complexity. Does mechanically activating the Goro circuit in intact larvae also lead to rapid seizures? Could the seizures be caused by channelrhodopsin axonal hyperactivation due to some other defect (the axonal diameter is reduced, so they might get hyper activated compared to controls) rather than by ephaptic coupling, since it seems these animals are pretty capable of coordinated motor behavior otherwise. If ephaptic coupling is the explanation for these seizures, the authors might be able to elicit the same motor response by exciting md4 while inhibiting Goro with inhibitory TRP or an Archhalo. This would be a convincing demonstration that peripheral sensory/motor "cross talk" is the cause of seizures.

4) If there is extensive sensory/motor ephaptic coupling caused by lack of wrapping glia, how is coordinated activity possible in the animal? They seem more normal than you would predict. Why does Goro stimulation lead to recurrent activation of the motor routine rather than just devolving into a seizure? Interpreting the causes of the behavioral defects reported in Figure 8 seem challenging and could use a more open discussion of other possible things at play.

5) Any idea why the animals die as pupae?

Reviewer #4:

Remarks to the Author:

In the original article, Kottmeier et al. present several fly models to study the role of enwrapping glia (EG) in the fly PNS. They first demonstrate that PNS EG in drosophila requires FGF-receptor. Second they show that innexins are necessary for the enwrapment of EG. Finally, they study the effect of PNS EG ablation in drosophila. Technically the study is well designed are there are only a few experiments necessary to consolidate the work (cf. major point). However, the manuscript feels disjointed and the collage of three different stories (FGF-r, innexins and PNS EG ablation). Several conclusions and correlations are not appropriate and need some rework (cf. major point). The manuscript would benefit if the authors would stress more the novelties of their results by showing how their work complement the existing literature on vertebrate non-myelinating glia.

Major

Figure 2 conclusion: "a block of FGF-receptor signaling impairs differentiation of the wrapping glial cells but the residual glial cell processes are still able to sustain neuronal function." is not fully demonstrated.

While *inx2* can still be expressed by subperineurial glial cells, how do authors determine that KD is efficient in Fig3G'? Staining does not look different from 3E'.

If the significant difference of 0.060m/s in NCV biologically relevant? Also I was not able to understand clearly how animal ablated of PNS EG are able to carry the information faster than their control counterpart.

There is a large portion of the discussion dedicated to comparing PNS EG results and PNS myelin in vertebrates. A better comparison and discussion would be made by comparing to PNS non-myelinating Schwann cells or olfactory ensheathing glia.

Minor

Authors refer in abstract to "brain". Nervous system is more appropriate.

There is a lonely m in the introduction next to ref 18.

"This suggests that wrapping glial cells perform additional tasks than just the acceleration of axon

potential propagation speed." - What suggests ?

In the first paragraph of the result there is a repetition of subsequently.

I do not understand what "the expression regime" is.

Figure 2A and B have two embedded legends.

Wrapping index (wi) is not define in the methods

The measurement of wrapping glia behavior is not defined in the methods (bending distribution, accumulated distance, number of stops, peristalsis frequency and efficacy)

Figure 4 conclusion: "In the absence of wrapping glial cells, axonal diameter is not growing as in control and thus, conduction speed is expected to be reduced." is not appropriate either because it is not demonstrated at this stage in the study.

Argumentation such as "Although ogre knockdown causes glial differentiation defects comparable to heartless suppression" is inappropriate as there is no direct evidence of the two being connected.

Dear reviewers,

we are thankful for all your comments which helped to further improve the manuscript. We also appreciate your positive view of our paper, and hope that after addressing all your comments and adding new and exciting additional results you find the paper now acceptable.

Our main points can be summarized as follows:

First of all, we analyzed larvae that expressed a dominant negative FGF-receptor in wrapping glia in much more detail. We now determined the axonal diameter in such nerves and surprisingly found that the axonal diameter is similarly affected as upon glial ablation. In addition, we determined the conduction velocity in peripheral axons, which is reduced similar as observed upon wrapping glia ablation.

We had already shown that FGF-receptor activity is needed for glial differentiation but mal-differentiated glial cells do not dramatically affect larval locomotion. We now performed a more detailed analysis of locomotor phenotypes. Wrapping glia ablated animals show a prominent coiling phenotype, which we now quantified. Whereas wild type larvae show a coiling index of about 0.1, wrapping glia ablated animals show a coiling index of 1.5 and animals expressing a dominant negative FGF-receptor display a coiling index of 0.8 (and animals with impaired gap junctional coupling show a coiling index of 0.7).

Reviewer 1 suggested to supplement the food with sphingosine. We are very thankful for this suggestion! Indeed, this rescued the axonal diameter as well as the wrapping index. More importantly the addition of sphingosine also rescued the coiling phenotype shown in larvae with suppressed FGF-receptor activity. Sphingosine supplementation, however, did not rescue the coiling phenotype in glia ablated larvae.

Moreover, we show that wrapping glia ablation but not suppression of FGF-receptor signaling in these cells changes the optogenetically triggered rolling response - although both paradigms have similar consequences for axonal conduction velocity. A more detailed list of the different changes to the manuscript follows in the below point to point reply.

We sincerely hope that the current version of the manuscript is now acceptable. With many thanks for your time in helping to get this work better,

Christian

Reviewer #1's comments to Authors:

In this manuscript Klämbt and colleagues report on the role of wrapping glia in the *Drosophila* using a combination of genetic, optogenetic and electrophysiological approaches. While wrapping glia tightly ensheath axons they do not form multilamellar membrane sheaths as the vertebrate oligodendroglial myelin sheaths. Here, the authors examine the hypothesis that wrapping glia, in homology to vertebrate myelin, contribute to conduction velocity and/or metabolic supply. The genetic approaches to ablate wrapping glia (based on expressing *Hid* and increasing specificity with the 90C03-Gal80 transgene combination), the whole-body imaging as well as the behavioral analyses are convincing. The questions are valid, timely and interesting. However, a number of claims made in the present manuscript version are speculative and the evidence that wrapping glia contributes to metabolic supply is absent.

Major points.

A) The main finding is that the authors (p. 4) "... identified behavioral changes that cannot be simply accounted to reduction action potential conduction speed". What is the evidence that reduced conduction velocity is insufficient? Which experiments are supporting those claims?

The reviewer is correct, and we apologize that we did not explain this better. We now added the results of additional experiments that clearly demonstrate this notion.

1) We analyzed the consequences of suppressing FGF-receptor activity specifically in the wrapping glia more intensively. This caused a similar reduction in axon caliber as noted upon ablation of the wrapping glia. This information is now added to Figure 2.

2) Suppression of FGF-receptor in wrapping glia causes a similar reduction in conduction velocity as noted upon ablation of the wrapping glia. This information is now added to Figure 7 where it is directly compared to the consequences of wrapping glia ablation.

3) Suppression of FGF-receptor signaling shows a small increase in the coiling index, which we defined as the number of full body turns in 30 seconds or 300 frames. This now added to Figure 2.

4) The optogenetic induction of rolling behavior is sensitive to wrapping glia ablation but is not affected by reduced wrapping glia differentiation as seen upon suppression of FGF-receptor activity. This information is now added to Figure 8

5) Knockdown of genes affecting ceramide synthesis also reduced conduction velocity (Ghosh et al 2015) but does not result in pronounced behavioral phenotypes (own data).

It is for example unclear how equation 1 (p. 17) is applied and used by the authors. Which numbers are thought to be used and what would be the predicted mean velocity change?

Equation 1 was used as follows. We set the experimentally determined the conduction speed in control animals and the corresponding radius, then we solved the equation to

$$v = \sqrt{a * \frac{K}{2RC}}$$

Thus, $\frac{v^2}{a} = \frac{K}{2RC} = \text{constant}$

When we insert for example $v_{mot} = 0.192$ and $a = 0.19$ we get the following value for the constant = 0.202. If we calculate the same for the conduction speed upon ablation of the wrapping glia we get a different value (0.104) suggesting that speed is more affected than expected. However, to avoid problems in understanding the logic we removed the section from the manuscript and just left the (as we think undisputed) notion that conduction velocity depends on axonal diameter.

Computational studies of myelin have shown that even the presence of a single membrane wrap, increasing the extracellular resistance and reducing axial leak, can support velocity increases. How do the authors distinguish between an electrical insulating effect and axon core diameter reduction?

We cannot clearly differentiate between the effects caused by a reduced axon diameter and the missing insulation. However, we would like to argue, that an axon either has an insulating glial sheet, or it is flanked by other axonal membranes (see Figure to the right, the blue axon, ax1, is insulated by other axons and their membrane, the yellow axon is insulated by glial and axonal membranes).

In terms of physical insulation there should be no difference between the two cases, except that there is some leakage in the paracellular spaces between axons. Moreover, glial cells never wrap all axons individually in a nerve, the wrapping index in wild type nerves is around 19 which means that the approximately 80 axons found in a peripheral nerve are clustered in 19 distinctly wrapped bundles.

B) There is no direct evidence for a role of metabolic supply. The conjecture that an average axon diameter reduction in the absence of wrapping glia is evidence for metabolic supply is correlative. Axon diameters are in part dependent on various neuron-glia interaction partners including Neuregulin. If the homology with vertebrate myelin is used the authors will need to demonstrate knocking out lactate or alanine uptake or other metabolic transporters mimics the findings and/or contribute to the phenotypes. Alternatively, experiments will need to demonstrate that morphological phenotype can be rescued, such as e.g. adding sphingosine to the diet (Ghosh et al., PloS Gen. 2013).

The reviewer is correct and the notion of metabolic supply of axons by glia is - at the current point - correlative. We have measured now more than 5.000 axons and found smaller diameters associated with both glial FGF-receptor knockdown or wrapping glia ablation. Metabolites first have to cross the blood-brain-barrier to reach wrapping glia and axons. Possibly amino acid or monocarboxylate transporters will be required. Drosophila harbors more than 20 monocarboxylate transporters and we have indeed done single knockdowns of individual members of this family but failed to detect any phenotypes. Even double or triple mutants that we have meanwhile generated in another project, failed to give conclusive results. Thus, we asked whether gap junctional coupling is needed for wrapping glia differentiation. The phenotypes shown in the paper are demonstrating that gap junctional coupling of glial cells is required. We do not know which solutes travel through gap junctions, but metabolites are

certainly possible. We changed the text accordingly and toned down the notion that we proved metabolic supply.

In addition, we performed the suggested rescue experiments and grew larvae expressing a dominant negative FGF-receptor with poorly differentiated wrapping glia on food supplemented with sphingosine. This treatment indeed resulted in an improved locomotor efficacy and also improved the wrapping index. These data are now added to Figure 2.

C) The terminology and use of electrophysiological readouts such as “ephaptic coupling” is confusing. Ephaptic coupling refers to the finite size and resistance of the extracellular space, causing electrical interference from an action potential in one fiber to the other. In the absence of wrapping glia and the increased extracellular space it may be even envisioned that electrical interactions will diminish.

The reviewer addresses an important point. In case of increased extracellular space, electrical interactions diminish, and in case of reduced space they will increase. In our electron microscopic analysis we detected smaller axons that were closer together. The spacing between axons is reduced but the extracellular space between axons is not changed.

On the other hand, as the authors also mention in the Discussion, “Ephaptic coupling can be beneficial during synchronous neuronal activity”. The authors interpret optogenetically evoked peripheral sensory input into the Goro circuit as seizures and evidence for ephaptic coupling. This is insufficient.

We agree with the reviewer and have changed the text to stress the following: Upon optogenetic stimulation of sensory neurons, control animals react with a well-defined motor response after a specific time delay (210 msec). Upon wrapping glia ablation, the animals react much faster (80 msec) with an unspecific motor response, although the conduction velocities are reduced by 30 % in wrapping glia ablated animals. If anything, this should slow down the reaction time.

The determined conduction velocity in wrapping glia ablated nerves is 0.133 m / sec for motor axons and 0.079 m / sec for sensory axons. For a 2 mm long abdominal nerve it thus takes 25.3 msec to transmit the signal from the sensory neuron to the CNS and 15 msec to send a signal from the CNS to the muscle. In addition, csChrimson

has relatively slow on-kinetics (8 msec to reach 90% opening in cell culture, (Klapoetke et al., 2014). This already accounts for about 50 of the 80 msec that we had determined as reaction time and indeed makes ephaptic coupling between sensory- and motor-axons likely. The strongest argument in favor for ephaptic coupling originates from the observation that optogenetic activation of a motor command neuron (the Goro neuron) triggers rolling in control larvae and in wrapping glia ablated larvae. However, in wrapping glia ablated larvae, the rolling orientation frequently changes and the response is longer than in control larvae. Thus, ectopic sensory input evoked by the strong motor activity sparks a new rolling response in a newly calculated direction which perpetuates the rolling response.

Nevertheless, since electrophysiological experiments unfortunately did not work, we changed the wording in our discussion, pointing out that ephaptic coupling is a possibility but requires further testing.

It is also unclear why this would inform us about “precision” of action potential propagation or whether this is evidence for “ephaptic coupling”. To demonstrate ephaptic coupling with electrophysiological means is indeed difficult. However, I would recommend the authors attempt to provide electrophysiological evidence for a role of wrapping glia in temporal precision, which is doable. The authors may want for example to record many trials with the extracellular responses (better using suction electrodes) and compare quantitative measures like jitter (i.e. the variation following a precisely triggered pulses).

We followed this advice and expressed the red light sensitive csChrimson in mdIV neurons using the *pickpocket* enhancer (*ppk-LexA*), and recorded the activity of muscle fiber 6 following optogenetic stimulation of the md4 neurons (supplementary Figure 1). Postsynaptic potential frequency increases with different intensity in control and wrapping glia ablated animals which is likely due to the reduced axonal conductance speed (see Figure 1 below). We did not detect evidence for a faster response that would be expected from ephaptic coupling but it must be stressed that we do not know the spatial relationship of sensory axons of the *ppk* positive sensory neurons and the single motor axon that projects to the M6 muscle.

This information is not mentioned in the current manuscript but we added the results of additional experiments. We now show that expression of the dominant negative FGF-receptor in wrapping glia causes a reduction of axonal diameter and a reduction of

conduction velocity - similar as observed in wrapping glia ablated larvae. However, the characteristic rolling response evoked upon optogenetic activation of either the mdIV neurons or the Goro neurons is only affected in animals with no wrapping glia. This demonstrates that the residual glial cell processes still present in nerves expressing a dominant negative FGF-receptor are still capable to guarantee neuronal signaling precision. In conclusion, we hope to convince the reviewer that wrapping glia contribute to both speed and precision of neuronal signaling. However, further work is needed to directly proof the existence of ephaptic coupling by individual axons in the abdominal nerve of the *Drosophila* larva.

Figure 1

Analysis of postsynaptic potential in the muscle fiber M6 upon optogenetic activation of the ppk-positive sensory neurons. A) Schematic view on the experimental setup. B) Individual postsynaptic potentials (PSP) over time in muscle fiber M6 in control and experimental animals. The genotypes are indicated. C) The mean PSP frequency over time increases faster in control than in experimental animals.

Minor points.

p. 3 The Introduction fails to provide a rigorous in-depth overview of wrapping glia in *Drosophila* but instead superficially describes aspects like “training of the brain” or the “allowance of dinosaurs”. This is interesting for an Opinion article or Review on myelin but has no place in an introduction into wrapping glia in the fly. This reviewer was missing an introduction about the genes used to enable wrapping glia. A more concise Introduction into those would be helpful. Introduce the genetics instead of a generic statement like “We established a tool”.

We are thankful for this suggestion and added more specific information on wrapping glia, which however increased the list of cited references. We also removed all dinosaurs...

p. 3 The sentence that “This suggests that wrapping glial cells perform additional tasks” I was not able to follow. How are the two preceding sentences about myelination in shrimps and copepods supporting this argument?

We are sorry, before submission of the first version we had deleted the relevant information and forgot to put it back again. This is now done. The paragraph now reads as follows:

Given the generally small size of invertebrates, no evolutionary pressure is expected to promote the development of very fast electrical conductance and thus myelin-like structures. Surprisingly, however, such myelin-like structures can be found in several invertebrates including shrimps, and copepods which due to their very small size of 200 μm length do not appear to require very fast nerve conduction velocity (Hartline, 2011; Hartline and Colman, 2007; Heuser and Doggenweiler, 1966; Roots and Lane, 1983; Wilson and Hartline, 2011a; Wilson and Hartline, 2011b; Xu and Terakawa, 1999). Indeed, swimming speed in copepods does not correlate with myelination (Buskey et al., 2017). This suggests that wrapping glial cells perform additional tasks than just the acceleration of axon potential propagation speed (Rey et al., 2020).

p.3. “Bundles m” ?

corrected

Figure 7. What do the multiple depolarizations in the extracellular recordings represent? A more detailed description would be useful. A particular concern is which waves are selected for calculating the temporal delay? In addition, how many trials are recorded? How robust are the differences?

Due to the differential recording technique, a single action potential of a motor neuron is recorded normally as a biphasic signal. Considering that a motor neuron must fire trains of action potentials to elicit contraction of a muscle, the multiple maxima (depolarizations) represent a series of action potentials fired by a single or multiple motor neurons. Moreover, the nerve recording includes the motor output of all motor neurons projecting through the nerve which innervate the whole musculature of one hemi-segment.

To determine the velocity in motor axons of control larvae a total of 222 spikes in 16 recordings were analyzed. To determine the velocity in larvae expressing a dominant negative FGF-receptor, 178 spikes in 10 recordings were analyzed, and 231 spikes in 14 recordings were analyzed for larvae with ablated wrapping glia. To determine the velocity in sensory axons of control larvae, 224 spikes in 16 recordings were analyzed. In wrapping glia ablated larvae 179 spikes in 14 recordings and in larvae expressing a dominant negative FGF-receptor 229 spikes in 10 recordings were analyzed. This information is now added

For the majority of the data reported in this study detailed information about supporting statistical approaches are lacking. Just one example: “upon ablation of wrapping glial cells axon diameters and the corresponding axons areas are reduced” (p. 9). No numbers, standard deviations nor supporting statistics are reported. Neither in the Figure legends. Proper scientific reporting for all statistical comparisons and claims needs to be included.

We are sorry, we added the information now.

In the methods “double-paired t-test” is mentioned which is unknown to this reviewer.

We are sorry for this mistake and corrected it to paired t-test and provided the software information.

Reviewer #2's comments to Authors:

In the current manuscript, the authors examine the role of wrapping glia in *Drosophila* larvae as a step towards myelinating glia in vertebrates. The authors generate a new genetic toolkit to manipulate wrapping glia specifically and show that reductions in their function cause only mild defects in larvae. In contrast, complete ablation of the wrapping glia leads to smaller diameter axons and a host of defective behaviors, including reduced motor and peripheral axon conduction speeds. The authors suggest their data support an increase in ephaptic coupling (increased electrical cross-talk between axons) based on a behavioral rolling response observed in a sensory to motor circuit, but this conclusion has the weakest support of the other claims. Overall, the authors generate a nice tool to study wrapping glia function as a precursor of the evolution of myelinating glia and report interesting functions for this glial population in regulating axonal size and action potential conduction velocity.

Minor Comments:

1) Images shown in Figure 2B and in Figure 3F/G are used to suggest that manipulating wrapping glia development makes the glia “thinner.” However, the EM images of these manipulations show the wrapping glia extending throughout the nerve as seen in controls. The only EM deficit appears to be failure of the glia to fully wrap around axons. Why does this ultrastructural change result in a thinner appearance by confocal microscopy? How to directly compare the EM and confocal images is a bit unclear. Indeed, understanding what is happening in the early manipulations of the wrapping glia before the authors ablate them is a bit tricky.

The reviewer is correct and we failed to properly explain the phenotype. We have now added the following:

This resulted in larvae with poorly differentiated wrapping glial cells (**Figure 2A,B**). In confocal microscopic views the wrapping glia appeared sometimes thinner as a single line within the axon bundle (compare arrows in **Figure 2A,B**). However, in other nerves or other sections of the same nerve, the wrapping glia appeared to cover a larger region of the nerve bundle. Following electron microscopic analysis, the disrupted differentiation of the wrapping glial cell can be clearly seen (**Figure 2C,D**). The wrapping glia fails to properly wrap around axons but is still able to extend processes through the nerve bundle (**Figure 2D**). Depending on the view angle, this might either appear as a thin or a broad line. The failure to individually wrap axons can be described using the wrapping index (w_i) (see Materials and Methods) (Matzat et al., 2015). In control larvae, the w_i is about 19 (Matzat et al., 2015). Upon expression of *htl^{DN}*, the w_i is about 7.5 (**Figure 2C,D,E**).

2) Figure 7 shows wrapping glia are important for regulating axonal conductance velocity. Whether this is directly caused by leaky conduction, or indirectly by failure to support axonal maturation is not clear. The authors favor reduced axon diameter as the explanation, but it seems there are other possibilities here that can't be ruled out. It's also difficult for me to determine how the data in Fig. 7B was generated – the two extracellular recordings don't always match up well, even in their presumed best sample traces they show – how are the authors sure their picking the same action potential to measure at the more distal electrode? This is some of the most critical data for the paper.

We have determined the conduction velocity for both motor as well as sensory axons. To differentiate the two modalities, we either utilized fictive crawling or sensory stimulation. In case of sensory stimulation, we can clearly elicit a train of spikes by mechanical irritation of the larval skin. In case of motor axons, we rely on autonomous activity of the central pattern generator which drives a process called fictive crawling. Here, trains of action potentials are sent to the muscle and the pattern of these spikes can be recognized in the extracellular recording. We decided against the delivery of an exogenous stimulus (e.g. current injection in the ventral nerve cord) since this rather artificial situation would employ all axons and thus would not allow a distinction between afferent and efferent fibers. We have added an additional explanatory sentence to the text.

3) Figure 8 is a nice experimental test of a larger sensory to motor circuit, but experimental interpretations seem open to some guesswork due to the complexity. Does mechanically activating the Goro circuit in intact larvae also lead to rapid seizures? Could the seizures be caused by channelrhodopsin axonal hyperactivation due to some other defect (the axonal diameter is reduced, so they might get hyper activated compared to controls) rather than by ephaptic coupling, since it seems these animals are pretty capable of coordinated motor behavior otherwise. If ephaptic coupling is the explanation for these seizures, the authors might be able to elicit the same motor response by exciting md4 while inhibiting Goro with inhibitory TRP or an Archhalo. This would be a convincing demonstration that peripheral sensory/motor “cross talk” is the cause of seizures.

The reviewer is very correct and silencing sensory neurons in the ablation paradigm would be an interesting experiment to do. However, it requires the generation of flies that harbor three independent binary activation systems to at the same time to manipulate csChrimson expression in some neurons, ablate wrapping glia cells and silence neuronal activity in other neurons. While the required *lexA* and *Gal4* transgenes are established in the lab, we would need to generate the corresponding

lines for the QF system - which would have been beyond the scope of this paper. Instead of silencing we instead attempted an additional ablation experiment where we tried to eliminate both wrapping glia and the mdIV neurons (or wrapping glia and Goro neurons). This could be done by recombining a *ppk-Gal4* or *goro-Gal4* insertion onto the *90C03-Gal80* chromosome used in our stocks. Unfortunately, we failed to obtain a *ppk-Gal4 90C03-Gal80* recombinant. We did obtain a *goro-Gal4 90C03-Gal80* recombinant but could not establish the stock that allows concomitant ablation of Goro neurons and wrapping glia.

4) If there is extensive sensory/motor ephaptic coupling caused by lack of wrapping glia, how is coordinated activity possible in the animal? They seem more normal than you would predict. Why does Goro stimulation lead to recurrent activation of the motor routine rather than just devolving into a seizure? Interpreting the causes of the behavioral defects reported in Figure 8 seem challenging and could use a more open discussion of other possible things at play.

The reviewer overlooked the point that coordinated movement of larvae is not possible. The regular peristaltic contraction waves are completely misshaped which might well be due to ephaptic coupling. We did not stress this in the paper because we did not want to interpret our data too much. Why does Goro activation not lead to seizures but rather to an almost normal appearing rolling behavior? This is a good point and we now discuss it more openly. In our view a likely explanation is the intensity of neuronal activation. We expanded the discussion and added the results of the additional experiments mentioned above to strengthen our argumentation.

5) Any idea why the animals die as pupae?

We have no idea why this could be. Metamorphosis requires a general and pronounced reorganization of the body and this might render an animal very sensitive to even tiny developmental defects that have occurred before.

Reviewer #4 (Remarks to the Author):

In the original article, Kottmeier et al. present several fly models to study the role of enwrapping glia (EG) in the fly PNS. They first demonstrate that PNS EG in drosophila requires FGF-receptor. Second they show that innexins are necessary for the enwrapment of EG. Finally, they study the effect of PNS EG ablation in drosophila. Technically the study is well designed are there are only a few experiments necessary to consolidate the work (cf. major point). However, the manuscript feels disjointed and the collage of three different stories (FGF-r, innexins and PNS EG ablation). Several conclusions and correlations are not appropriate and need some rework (cf. major point). The manuscript would benefit if the authors would stress more the novelties of their results by showing how their work complement the existing literature on vertebrate non-myelinating glia.

Major

Figure 2 conclusion: “a block of FGF-receptor signaling impairs differentiation of the wrapping glial cells but the residual glial cell processes are still able to sustain neuronal function.” is not fully demonstrated.

We had previously only conducted a behavioral analysis focusing on accumulated distance and bending frequency. As the reviewer correctly identified, this indeed does not inform the reader about specific neuronal function. To address this point we have now added the following new experimental data:

We have determined the size of the different axons in animals expressing a dominant negative FGF-receptor (more than 2000 axons counted) and found a similar reduction in size as noted for the ablation of the wrapping glia.

We next determined the conduction velocity as we did following wrapping glia ablation. As expected from the reduced axonal diameter we found a similar impairment of axonal conduction velocity.

We then analyzed the behavioral consequences of wrapping glial specific expression of a dominant FGF-receptor in much more depth. We found that the coiling behavior, which is characteristic for larvae that specifically lack the wrapping glia can be found in such animals as well. Albeit at much lower rates - which explains why we did not identify this phenotype in the first place.

Finally, we performed additional behavioral experiments. Control larvae show a characteristic rolling response upon optogenetic activation of either the mdIV neurons or the Goro neurons. This response is differentially affected upon ablation of the

wrapping glia. Interestingly, expression of a dominant negative FGF-receptor does not cause similar deviations of the rolling response - although the nerve conduction velocity is affected similarly in both paradigms. This demonstrates that the residual glial cell processes still present in nerves expressing a dominant negative FGF-receptor are still capable to block ephaptic coupling.

While *inx2* can still be expressed by subperineurial glial cells, how do authors determine that KD is efficient in Fig3G'? Staining does not look different from 3E'.

We agree the quality of the data shown in the previous Figure 3E do not allow this conclusion. We have revised Figure 3 and removed the part addressing the glial subtype specific expression of innexins from the text (and the Figure). Instead we added new behavioral analysis looking at the coiling phenotype.

If the significant difference of 0.060m/s in NCV biologically relevant? Also I was not able to understand clearly how animal ablated of PNS EG are able to carry the information faster than their control counterpart.

We are sorry that we failed to properly explain this and improved this in the revised version.

In the first version of our manuscript, we had only characterized animals lacking wrapping glia. They showed reduced conductance velocity in the peripheral nervous system AND a larval locomotor phenotype. Now we characterized larvae expressing a dominant negative FGF-receptor in wrapping glia. They have similarly reduced axonal diameters and show a reduced conductance speed which is very similar to what we observed following wrapping glia ablation. In stark contrast, however, such larvae have only very mild behavioral phenotypes.

Now we can conclude that the reduced conductance velocity has no big impact on the general locomotor pattern of the *Drosophila* larva.

The second aspect of the question deals with possible ephaptic coupling which we failed to properly explain in the last version. Here information is conducted slower - but due to electrical crosstalk between sensory and motor axons the animal reacts faster with a distinct behavior and shows seizures. The speed of the reaction could be explained by the fact that the optogenetically activated sensory neurons directly

activate motor axons. A similar case is found when we activated the Goro neurons which normally trigger the rolling response. Here ablation of wrapping glia causes increased rolling in changing orientations (clockwise and counter clockwise). Again, this could be explained by ephaptic coupling where optogenetically activated motor neurons activate sensory neurons. The sensory input then (a) sustains the rolling response and (b) causes constant reorientation.

There is a large portion of the discussion dedicated to comparing PNS EG results and PNS myelin in vertebrates. A better comparison and discussion would be made by comparing to PNS non-myelinating Schwann cells or olfactory ensheathing glia.

We are thankful for this suggestion and invested in the discussion comparing *Drosophila* wrapping glial cells to non-myelinating Schwann cells. This indeed generated a better focus and the findings mentioned in the newly mentioned papers strengthened our argumentation. We added:

In the mammalian peripheral nervous system differential axonal caliber has long been recognized as an important denominator of myelination (Matthews, 1968; Peters et al., 1991). Whereas large caliber axons are associated with myelinating Schwann cells, small caliber axons are associated with non-myelinating Schwann cells that form Remak fibers (Fledrich et al., 2019; Jessen and Mirsky, 2005; Napoli et al., 2012). Many of the axons in Remak bundles are sensory C-fibers which transmit pain information (Harty and Monk, 2017; Maarbjerg et al., 2017). Differentiation of the non-myelinating Schwann cells is in part regulated by GABA-B-receptor signaling (Faroni et al., 2014). Schwann cell specific knock-out mouse mutants show an increase in the number of small unmyelinated fibers. Interestingly, such mice show an increased sensitivity to pain (hyperalgesia) and, as also seen upon deletion of the LDL receptor-related protein-1, show pain reactions to stimuli that normally do not elicit pain (allodynia) (Faroni et al., 2014; Orita et al., 2013). Neuropathic pain due to poorly differentiated non-myelinating Schwann cells might be indicative for ephaptic coupling of closely contacting axons similar to what we reported here.

Non-myelinating Schwann cells, which share a common lineage with myelinating Schwann cells (Hurley et al., 2007; Ma et al., 2018; Stierli et al., 2018), are morphologically similar to the wrapping glia that engulf peripheral axons in *Drosophila*

(Harty and Monk, 2017; Matzat et al., 2015; Stork et al., 2008). Moreover, similar molecular processes appear to operate in flies and mammals. In mice, deletion of FGF-receptor signaling specifically in non-myelinating Schwann cells leads to abnormally and smaller appearing differentiated axons (Furusho et al., 2009). Likewise, differentiation of non-myelinating Schwann cells in mice requires neuropathy target esterase while glial loss of the *Drosophila* homolog *swiss cheese* causes incomplete glial wrapping as well as locomotion defects (Dutta et al., 2016; McFerrin et al., 2017).

Minor

Authors refer in abstract to “brain”. Nervous system is more appropriate.

Corrected

There is a lonely m in the introduction next to ref 18.

Deleted

“This suggests that wrapping glial cells perform additional tasks than just the acceleration of axon potential propagation speed.” - What suggests ?

We are sorry, a sentence was missing here and now has been added:

Indeed, swimming speed in copepods does not correlate with myelination (Buskey et al., 2017). This suggests that wrapping glial cells perform additional tasks than just the acceleration of axon potential propagation speed.

In the first paragraph of the result there is a repetition of subsequently.

Corrected.

I do not understand what “the expression regime” is.

We deleted the words.

Figure 2A and B have two embedded legends.

The Figure is now corrected.

Wrapping index (wi) is not defined in the methods.

We added the following to Materials and Methods:

To calculate the wrapping index the number of individually wrapped axons or axon fascicles is put into relation to the number of all axons. A wrapping index of 100 % implies, that every single axon of the nerve is individually wrapped. All nerves that contained less than 76 or more than 82 axons were not included in the statistical analysis. Significance was determined using the Student's t-test.

The measurement of wrapping glia behavior is not defined in the methods (bending distribution, accumulated distance, number of stops, peristalsis frequency and efficacy)

We added to the Materials and Methods section:

Output files were statistically analyzed with MatLab (MathWorks, Mann Whitney Rank Sum tests). Accumulated distance describes the total length of larval trajectories per minute. Distance to origin describes the distance the larvae moved away from the spot they were placed normalized per minute. A stop is defined as no movement for at least 5 frames. The number of stops per minute and animal is given. A head bend is defined when a larva shows a bending angle of at least 20° for at least 5 frames. The number of head bends per 10 sec is given. In bending distribution plots all deviations from an 180° body axis are show. 180° indicates no turn and straight appearance. Turns are defined as a stop lasting at least 4 frames followed by a body bending of at least 20° and subsequent forward locomotion towards the new direction for at least 20 frames. Peristalsis frequency gives the number of body contractions as defined by rhythmic alterations in body size per 10 seconds. Peristalsis efficacy defines the difference between maximal and minimal body size, e.g. how much a larva can contract or extend which in wild type is about 10 % of the body size.

Figure 4 conclusion: "In the absence of wrapping glial cells, axonal diameter is not growing as in control and thus, conduction speed is expected to be reduced." is not appropriate either because it is not demonstrated at this stage in the study.

We deleted the sentence.

Argumentation such as "Although ocre knockdown causes glial differentiation defects comparable to heartless suppression" is inappropriate as there is no direct evidence of the two being connected.

We agree and have changed the sentence to:

ogre knockdown causes glial differentiation defects and mild locomotion defects in third instar larvae.

References:

- Buskey, E. J., Strickler, J. R., Bradley, C. J., Hartline, D. K. and Lenz, P. H. (2017). Escapes in copepods: comparison between myelinate and amyelinate species. *Journal of Experimental Biology* 220, 754–758.
- Dutta, S., Rieche, F., Eckl, N., Duch, C. and Kretzschmar, D. (2016). Glial expression of Swiss cheese (SWS), the *Drosophila* orthologue of neuropathy target esterase (NTE), is required for neuronal ensheathment and function. *Disease Models & Mechanisms* 9, 283–294.
- Faroni, A., Castelnovo, L. F., Procacci, P., Caffino, L., Fumagalli, F., Melfi, S., Gambarotta, G., Bettler, B., Wrabetz, L. and Magnaghi, V. (2014). Deletion of GABA-B receptor in Schwann cells regulates remak bundles and small nociceptive C-fibers. *Glia* 62, 548–565.
- Fledrich, R., Kungl, T., Nave, K.-A. and Stassart, R. M. (2019). Axo-glial interdependence in peripheral nerve development. *Development* 146.
- Furusho, M., Dupree, J. L., Bryant, M. and Bansal, R. (2009). Disruption of fibroblast growth factor receptor signaling in nonmyelinating Schwann cells causes sensory axonal neuropathy and impairment of thermal pain sensitivity. *Journal of Neuroscience* 29, 1608–1614.
- Hartline, D. K. (2011). The evolutionary origins of glia. *Glia* 59, 1215–1236.
- Hartline, D. K. and Colman, D. R. (2007). Rapid conduction and the evolution of giant axons and myelinated fibers. *Curr Biol* 17, R29–35.
- Harty, B. L. and Monk, K. R. (2017). Unwrapping the unappreciated: recent progress in Remak Schwann cell biology. *Curr Opin Neurobiol* 47, 131–137.
- Heuser, J. E. and Doggenweiler, C. F. (1966). The fine structural organization of nerve fibers, sheaths, and glial cells in the prawn, *Palaemonetes vulgaris*. *J Cell Biol* 30, 381–403.
- Hurley, P. A., Crook, J. M. and Shepherd, R. K. (2007). Schwann cells revert to non-myelinating phenotypes in the deafened rat cochlea. *Eur. J. Neurosci.* 26, 1813–1821.
- Jessen, K. R. and Mirsky, R. (2005). The origin and development of glial cells in peripheral nerves. *Nat Rev Neurosci* 6, 671–682.
- Ma, D., Wang, B., Zawadzka, M., Gonzalez, G., Wu, Z., Yu, B., Rawlins, E. L., Franklin, R. J. M. and Zhao, C. (2018). A Subpopulation of Foxj1-Expressing, Nonmyelinating Schwann Cells of the Peripheral Nervous System Contribute to Schwann Cell Remyelination in the Central Nervous System. *Journal of Neuroscience* 38, 9228–9239.
- Maarbjerg, S., Di Stefano, G., Bendtsen, L. and Cruccu, G. (2017). Trigeminal neuralgia - diagnosis and treatment. *Cephalalgia* 37, 648–657.
- Matthews, M. A. (1968). An electron microscopic study of the relationship between axon diameter and the initiation of myelin production in the peripheral nervous system. *Anat. Rec.* 161, 337–351.
- Matzat, T., Sieglitz, F., Kottmeier, R., Babatz, F., Engelen, D. and Klämbt, C. (2015). Axonal wrapping in the *Drosophila* PNS is controlled by glia-derived neuregulin homolog Vein. *Development* 142, 1336–1345.
- McFerrin, J., Patton, B. L., Sunderhaus, E. R. and Kretzschmar, D. (2017). NTE/PNPLA6 is expressed in mature Schwann cells and is required for glial ensheathment of Remak fibers. *Glia* 65, 804–816.
- Napoli, I., Noon, L. A., Ribeiro, S., Kerai, A. P., Parrinello, S., Rosenberg, L. H., Collins, M. J., Harrisingsh, M. C., White, I. J., Woodhoo, A., et al. (2012). A Central Role for the ERK-Signaling Pathway in Controlling Schwann Cell Plasticity and Peripheral Nerve Regeneration In Vivo. *Neuron* 73, 729–742.

- Orita, S., Henry, K., Mantuano, E., Yamauchi, K., De Corato, A., Ishikawa, T., Feltri, M. L., Wrabetz, L., Gaultier, A., Pollack, M., et al. (2013). Schwann cell LRP1 regulates remak bundle ultrastructure and axonal interactions to prevent neuropathic pain. *Journal of Neuroscience* 33, 5590–5602.
- Peters, A., Palay, S. and Webster, H. (1991). *The Fine Structure of the Nervous System*. Third Edition. New York, Oxford: Oxford University Press.
- Rey, S., Zalc, B. and Klämbt, C. (2020). Evolution of glial wrapping: A new hypothesis. *Devel Neurobio* 34, 5089–11.
- Roots, B. I. and Lane, N. J. (1983). Myelinating glia of earthworm giant axons: thermally induced intramembranous changes. *Tissue Cell* 15, 695–709.
- Stierli, S., Napoli, I., White, I. J., Cattin, A.-L., Monteza Cabrejos, A., Garcia Calavia, N., Malong, L., Ribeiro, S., Nihouarn, J., Williams, R., et al. (2018). The regulation of the homeostasis and regeneration of peripheral nerve is distinct from the CNS and independent of a stem cell population. *Development* 145.
- Stork, T., Engelen, D., Krudewig, A., Silies, M., Bainton, R. J. and Klämbt, C. (2008). Organization and function of the blood-brain barrier in *Drosophila*. *Journal of Neuroscience* 28, 587–597.
- Wilson, C. H. and Hartline, D. K. (2011a). The novel organization and development of copepod myelin. I. Ontogeny. *J. Comp. Neurol.*
- Wilson, C. H. and Hartline, D. K. (2011b). Novel organization and development of copepod myelin. ii. nonglial origin. *J. Comp. Neurol.* 519, 3281–3305.
- Xu, K. and Terakawa, S. (1999). Fenestration nodes and the wide submyelinic space form the basis for the unusually fast impulse conduction of shrimp myelinated axons. *J Exp Biol* 202, 1979–1989.

Reviewers' Comments:

Reviewer #1:

Remarks to the Author:

The authors have satisfactorily addressed most of my previous concerns and the manuscript substantially improved. The rescue experiments with sphingosine strengthen the evidence for a role of metabolic supply and the experimental data is described in greater detail. The figures the authors provided in the rebuttal were useful in the evaluation. There are some issues pending in writing and data presentation.

- The authors appropriately addressed the issue of ephaptic coupling in the rebuttal. They write: "We changed the wording in our discussion, pointing out that ephaptic coupling is a possibility but requires further testing." The experiments shown in Figure 1 (in the rebuttal) are nice but as the authors also acknowledge, it is not providing evidence for coupling. For this reason, it is somewhat surprising to learn that the manuscript still presents ephaptic coupling as a core hypothesis. Line 27 (abstract): "whereas signaling precision requires a block of electrical crosstalk between axons, known as ephaptic coupling. It is currently not well understood how speed and signaling precision are regulated." Two comments. 1) Spike precision is much more complex dependent on a wide range of mechanisms including membrane properties, sodium channels, channel kinetics, ephaptic interactions etc. 2) In the absence of formal evidence for spike-to-spike coupling between sensory and motor axons the claims about ephaptic cannot be made. The behavioural responses are interesting and an important observation in the context of wrapping glia, but they provide only a remote readout of neural processing. The claims should be toned down throughout the manuscript and most notably in the abstract. The abstract should provide a testable research question concerning glial wrapping. Since the length of the abstract is too long (150 words is max.) the ephaptic concepts could be moved into the Discussion.

- The authors provide now the supporting quantification for experiments, but statistical analyses are a concern. 1) It is unclear whether the data in Figures 6 to 8, with more than two groups, have been analyzed with ANOVA followed by post-hoc tests. At line 671 it reads "Significance was determined using the Student's t-test". If so, this is invalid. 2) Percentages cannot be analyzed with parametric test, please transform or better use absolute numbers, testing for normality to decide on parametric or non-parametric tests. 3) In line 382 it seems that a P value of 0.074 is considered significant? 4) Data are sometimes presented as violin or box plots with medians and appear chaotic in style and content. For example, were outliers left out of the analysis? 5) Decimal places are inconsistent indicated with periods or commas in both text and figures (e.g. Figure 4I versus Figure 4J). Taken together, this is not publishable. The manuscript should be carefully scrutinized and be proofread to adhere to standards of statistical reporting in the field.

- 'Neuronal conductance' and 'conductance velocity' are not equivalent terms for conduction velocity. Conductance is the property of the membrane measured in units of siemens whereas conductivity refers to ability of nerves or any material to propagate signals. Please correct throughout.

- The authors make a prediction for a minimum delay of transmission from sensory stimulus to motor behavior (lines 550-560). This number should also take into account the temporal delays caused by CNS circuitry in the brain. Each connection may easily cause a 10 ms lag (and thus 40 ms with 4 synapses, based on Fig. 8A).

Reviewer #2:

Remarks to the Author:

I'm happy with the additional data and improvements to conduction velocity measurements. In addition, the authors have toned down some of the stronger claims that were not convincingly

demonstrated, such as ephaptic coupling. The only remaining suggestion is to tone that comment down in the abstract as well -- something like "... the presence of wrapping glia which may ensure a block of ephaptic coupling" rather than the current "ensure" statement.

Reviewer #4:

Remarks to the Author:

The authors have addressed all comments well.

Reviewer #1 (Remarks to the Author):

The authors have satisfactorily addressed most of my previous concerns and the manuscript substantially improved. The rescue experiments with sphingosine strengthen the evidence for a role of metabolic supply and the experimental data is described in greater detail. The figures the authors provided in the rebuttal were useful in the evaluation. There are some issues pending in writing and data presentation.

- The authors appropriately addressed the issue of ephaptic coupling in the rebuttal. They write: "We changed the wording in our discussion, pointing out that ephaptic coupling is a possibility but requires further testing." The experiments shown in Figure 1 (in the rebuttal) are nice but as the authors also acknowledge, it is not providing evidence for coupling. For this reason, it is somewhat surprising to learn that the manuscript still presents ephaptic coupling as a core hypothesis. Line 27 (abstract): "whereas signaling precision requires a block of electrical crosstalk between axons, known as ephaptic coupling. It is currently not well understood how speed and signaling precision are regulated." Two comments. 1) Spike precision is much more complex dependent on a wide range of mechanisms including membrane properties, sodium channels, channel kinetics, ephaptic interactions etc. 2) In the absence of formal evidence for spike-to-spike coupling between sensory and motor axons the claims about ephaptic cannot be made. The behavioural responses are interesting and an important observation in the context of wrapping glia, but they provide only a remote readout of neural processing. The claims should be toned down throughout the manuscript and most notably in the abstract. The abstract should provide a testable research question concerning glial wrapping. Since the length of the abstract is too long (150 words is max.) the ephaptic concepts could be moved into the Discussion.

We have shortened the abstract and removed ephaptic coupling as a possibility to explain our data. This is now only mentioned in more detail in the discussion.

- The authors provide now the supporting quantification for experiments, but statistical analyses are a concern.

1) It is unclear whether the data in Figures 6 to 8, with more than two groups, have been analyzed with ANOVA followed by post-hoc tests. At line 671 it reads "Significance was determined using the Student's t-test". If so, this is invalid.

We are sorry for the misunderstanding. In no experiment shown in Figures 6 to 8 more than two groups were statistically compared. If the data was distributed normally, a parametric t-test was used for analysis. If not, the Wilcoxon rank-sum / Mann-Whitney U test was used.

This is now explicitly mentioned in the text.

Line 671 refers to the statistical analysis of wrapping indices. Due to normally distributed data, the parametric t-test was used.

2) Percentages cannot be analyzed with parametric test, please transform or better use absolute numbers, testing for normality to decide on parametric or non-parametric tests.

Again, we are sorry that we failed to properly explain this, which we have now done better in the Methods section. Data was first checked for normality and the statistical analysis was chosen accordingly. The coiling index was found to have a non-parametric distribution (Shapiro Wilk test) and thus a non-parametrical Wilcoxon rank-sum test was performed.

3) In line 382 it seems that a P value of 0.074 is considered significant?

Yes, the reviewer is absolutely correct. This is our mistake. The correct p-value is 0.0474, it was correctly mentioned in the legend to Figure 6 but in the main text a typo happened. We apologize and changed the value in the text.

4) Data are sometimes presented as violin or box plots with medians and appear chaotic in style and content. For example, were outliers left out of the analysis?

Most of the data are presented as box plots. Only in case of the bending distributions the violin plots were preferred since they provide a possibility to see the differences between the genotypes in an easier way.

For the electrophysiological experiments outliers were determined using Grubb's test and removed from the analysis.

For the behavioral analyses, no outliers were excluded from the statistical analyses.

5) Decimal places are inconsistent indicated with periods or commas in both text and figures (e.g. Figure 4I versus Figure 4J).

The reviewer is correct. We have now carefully looked at all Figures and replaced the commas with periods where necessary.

Taken together, this is not publishable. The manuscript should be carefully scrutinized and be proofread to adhere to standards of statistical reporting in the field.

We hope that we have now addressed all issues.

- 'Neuronal conductance' and 'conductance velocity' are not equivalent terms for conduction velocity. Conductance is the property of the membrane measured in units of

siemens whereas conductivity refers to ability of nerves or any material to propagate signals. Please correct throughout.

We corrected this throughout the manuscript.

- The authors make a prediction for a minimum delay of transmission from sensory stimulus to motor behavior (lines 550-560). This number should also take into account the temporal delays cause by CNS circuitry in the brain. Each connection may easily cause a 10 ms lag (and thus 40 ms with 4 synapses, based on Fig. 8A).

We are thankful for this suggestion and added a corresponding sentence in the discussion.

This even more favors our idea of ephaptic coupling between sensory and motor neurons.

Reviewer #2 (Remarks to the Author):

I'm happy with the additional data and improvements to conduction velocity measurements. In addition, the authors have toned down some of the stronger claims that were not convincingly demonstrated, such as ephaptic coupling. The only remaining suggestion is to tone that comment down in the abstract as well -- something like " ... the presence of wrapping glia which may ensure a block of ephaptic coupling" rather than the current "ensure" statement.

We have shortened the abstract and removed ephaptic coupling as a possibility to explain our data. This is now only mentioned in more detail in the discussion.

Reviewer #4 (Remarks to the Author):

The authors have addressed all comments well.